# A pentameric protein ring with novel architecture is required for herpesviral packaging

Allison L Didychuk[1], Stephanie N Gates[2,3,4], Matthew R Gardner[1], Lisa M Strong[2], Andreas Martin[2,3,4], Britt A Glaunsinger[1,2,3,4]*

[1]Department of Plant and Microbial Biology, University of California, Berkeley, Berkeley, United States; [2]Department of Molecular and Cell Biology, University of California, Berkeley, Berkeley, United States; [3]California Institute for Quantitative Biosciences, University of California, Berkeley, Berkeley, United States; [4]Howard Hughes Medical Institute, University of California, Berkeley, Berkeley, United States

**Abstract** Genome packaging in large double-stranded DNA viruses requires a powerful molecular motor to force the viral genome into nascent capsids, which involves essential accessory factors that are poorly understood. Here, we present structures of two such accessory factors from the oncogenic herpesviruses Kaposi's sarcoma-associated herpesvirus (KSHV; ORF68) and Epstein–Barr virus (EBV; BFLF1). These homologous proteins form highly similar homopentameric rings with a positively charged central channel that binds double-stranded DNA. Mutation of individual positively charged residues within but not outside the channel ablates DNA binding, and in the context of KSHV infection, these mutants fail to package the viral genome or produce progeny virions. Thus, we propose a model in which ORF68 facilitates the transfer of newly replicated viral genomes to the packaging motor.

*For correspondence:
glaunsinger@berkeley.edu

## Introduction

Herpesviruses are large double-stranded DNA viruses that cause a variety of diseases in humans. The ability of herpesviruses to efficiently evade the immune system and establish latency, coupled with few available treatments and vaccines, means that nearly all adults in the world harbor at least one of the nine human herpesviruses. Herpes simplex virus type 1 (HSV-1) is an alphaherpesvirus that causes cold sores and genital sores. Human cytomegalovirus (HCMV) is a betaherpesvirus that can cause mononucleosis and congenital birth defects. The human gamma-herpesviruses Kaposi's sarcoma-associated virus (KSHV) and Epstein–Barr virus (EBV) are oncogenic viruses, causing cancers such as primary effusion lymphoma and Kaposi's sarcoma (in the case of KSHV).

Despite 400 million years of evolution separating the human herpesviruses, several core pathways in replication are conserved (*McGeoch et al., 2006*). Near the end of the lytic cycle, herpesviruses replicate their genome as a head-to-tail concatemer of linked genomes separated by terminal repeats. Cleavage to produce a unit-length genome is intimately tied to packaging and occurs only after that genome is successfully transferred into a capsid. DNA packaging in tailed bacteriophages is thought to be mechanistically similar to that of herpesviruses (*Rixon and Schmid, 2014*). Despite infecting hosts in different kingdoms, both groups of viruses use an icosahedral capsid and an architecturally similar portal protein through which DNA is packaged (*Rixon and Schmid, 2014*; *Dedeo et al., 2019*). Furthermore, both depend on a 'terminase' motor responsible for packaging and cleavage of the genome. The large subunit of the terminase is the most conserved gene across the herpesviruses and possesses sequence and structural similarity to phage terminases, supporting the hypothesis that packaging occurs through an evolutionarily ancient mechanism (*Rixon and*

*Schmid, 2014*; *Nadal et al., 2010*; *Selvarajan Sigamani et al., 2013*). Packaging minimally requires recognition of the viral genome by the terminase, docking of the terminase-bound genome at the portal of a nascent capsid, translocation of the genome into the capsid by the terminase, and cleavage to release the remaining unpackaged concatemeric genome.

Cleavage and packaging in the herpesviruses, best studied in HSV-1, requires six conserved proteins in addition to the nascent capsid and concatemeric genome (*Heming et al., 2017*): HSV-1 UL6, UL15, UL17, UL28, UL32, and UL33. Three of these proteins (UL15/UL28/UL33) form the terminase motor, and the portal protein is composed of a dodecamer of UL6 (*Newcomb et al., 2001*; *Patel et al., 1996*). UL17 encodes a capsid vertex-specific protein important for stabilizing the capsid (*Gong et al., 2019*; *Grzesik et al., 2017*; *Liu et al., 2019*). In contrast, despite observations that UL32 and its homologs in HCMV (UL52) and KSHV (ORF68) are essential for production of packaged virions, their function in packaging remains unknown (*Albright et al., 2015*; *Gardner and Glaunsinger, 2018*; *Lamberti and Weller, 1998*; *Borst et al., 2008*). Phages lack an identifiable homolog of UL32 or ORF68, suggesting that an additional level of complexity exists in herpesvirus packaging.

Here, we applied a combination of structural biology and biochemistry to better define the role of the essential accessory protein ORF68 in KSHV packaging. We reveal the structure of KSHV ORF68 and its homolog in EBV (BFLF1), which adopt a novel fold and assemble into homopentameric rings. The similarity of these structures, combined with negative stain electron microscopy of homologs from HSV-1 and HCMV, suggests that this topology is conserved across the *Herpesviridae*. The central channel of ORF68 is lined with positively charged residues that are necessary for nucleic acid binding and production of infectious virions. We hypothesize that the viral genome is threaded through the ORF68 ring, and that ORF68 acts as a scaffold on which the terminase assembles for genome packaging.

## Results

### ORF68 forms a homopentameric ring

To structurally characterize KSHV ORF68, we purified the full-length protein from transiently transfected HEK293T cells. In agreement with our prior observation that ORF68 forms a multimer in vitro (*Gardner and Glaunsinger, 2018*), negative stain electron microscopy (EM) revealed rings comprised of five subunits (*Figure 1—figure supplement 1a*). A cryo-EM reconstruction of the pentamer was determined to 3.37 Å, from which an alanine backbone model was built (*Figure 1—figure supplement 1b*). However, as ORF68 bears no sequence homology to proteins outside of the *Herpesviridae*, de novo modeling was challenging. We obtained crystals of ORF68 that diffracted X-rays to a maximum resolution of 2.22 Å and used molecular replacement with the initial cryo-EM model to determine the structure of ORF68. Representative diffraction data are presented in *Figure 1—figure supplement 2*, and data collection and structure refinement statistics are listed in *Supplementary file 1* (cryo-EM) and *Supplementary file 2* (X-ray). The majority of the protein could be built in the X-ray structure, except for two disordered loops (residues 64–67 and 241–262), and a region in the central channel (residues 169–172 and 179–188). These regions were similarly disordered in the cryo-EM maps, suggesting inherent flexibility.

ORF68 forms a homopentameric ring with a diameter of ~120 Å (*Figure 1a*). Each monomer contains three zinc fingers and several bundles of α-helices connected by loops. A DALI search (*Holm and Rosenström, 2010*) identified proteins that are similarly α-helical in nature, but none with globally similar structures, suggesting that ORF68 adopts a novel fold. On the 'top' face of the ring lies a short semi-structured loop (residues P27–N36), which represents a region where large insertions are observed in alpha- and beta-herpesviruses homologs (*Figure 1—figure supplement 3*). The central channel is constricted toward the top of the ring, with a width of ~25 Å, and widens to ~45 Å at the bottom (*Figure 1a*). Two segments of each ORF68 monomer directly face this central channel: residues 167–188 and 435–451. Residues 435–451 form an α-helix that is anchored by H452, which coordinates a Zn ion at the bottom of the ring (*Figure 1b*). Interestingly, residues 167–188 are largely disordered, despite being anchored by C191 and C192 that participate in coordination of the same Zn ion.

ORF68 is stable as a pentamer even at high concentrations of monovalent salt (*Figure 1—figure supplement 4*). The subunit interface includes ~1200 Å$^2$ of buried surface area, consisting largely of

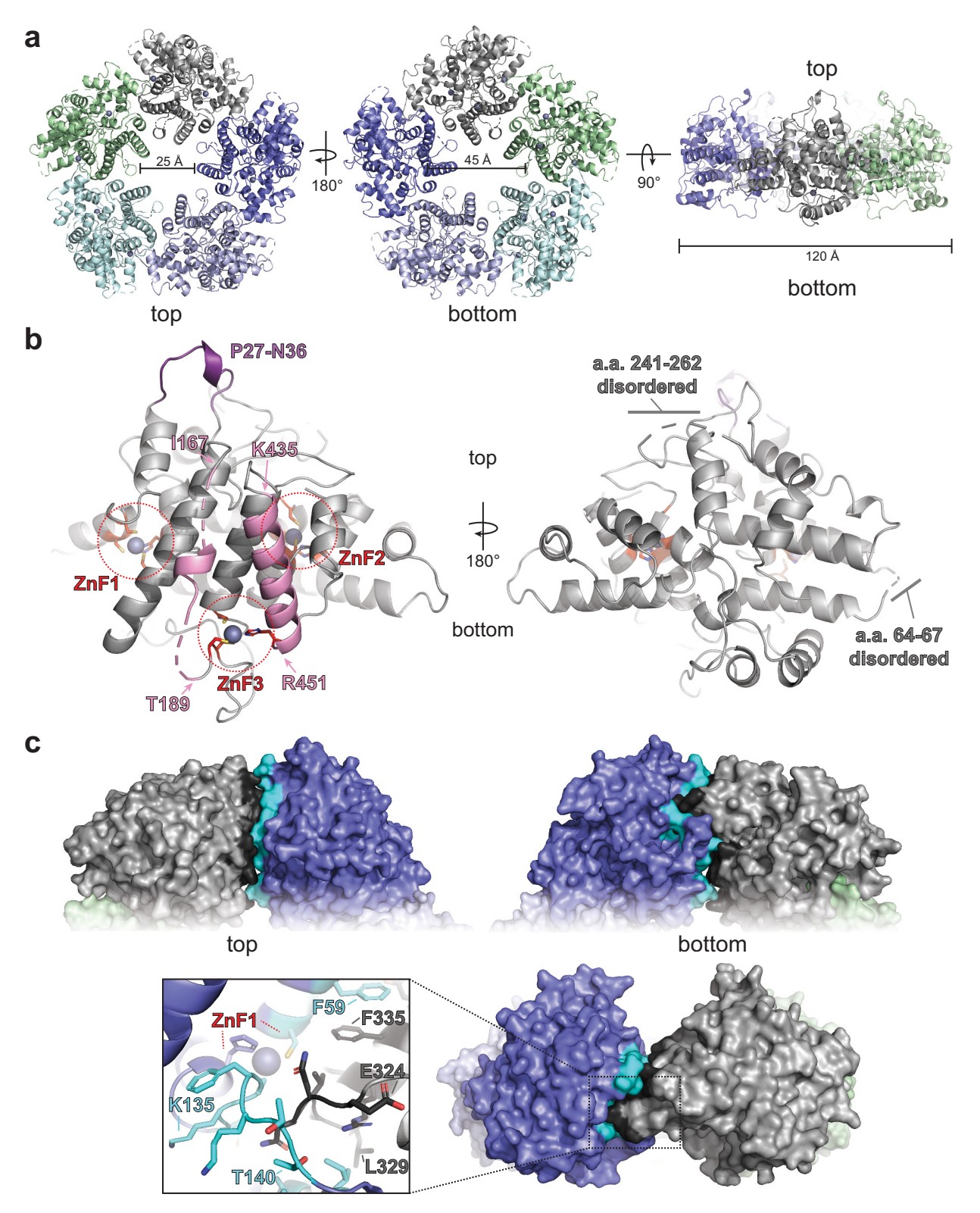

**Figure 1.** ORF68 forms a homopentameric ring. (a) View from the top, bottom, and side of the ORF68 crystal structure. The central channel size and overall diameter are highlighted. (b) Each monomer of ORF68 contains three zinc fingers (ZnF; coordinating residues shown in red sticks); Zn$^{+2}$ shown in gray spheres. Residues 167–188 and 435–451 (pink) span the central channel. Residues 435–451 form an α-helix, whereas residues 177–188 are largely disordered (highlighted in pink). Both regions are anchored by ZnF3. The 'top' of the ring has a semi-structured loop consisting of residues P27-N36

*Figure 1 continued on next page*

*Figure 1 continued*

(purple). Residues 241–262 and 64–67 are disordered. (c) Subunit interface within the ORF68 pentamer, with the monomer–monomer interface of the gray monomer highlighted in black and the interface on the blue monomer highlighted in cyan. ZnF1 is near the interface; a loop consisting of residues E324-L329 extends into the adjacent monomer. Residues F59 and F335 from adjacent monomers stack.

The online version of this article includes the following figure supplement(s) for figure 1:

**Figure supplement 1.** EM data analysis for ORF68.

**Figure supplement 2.** Representative electron densities and X-ray structural details of ORF68.

**Figure supplement 3.** Sequence alignment of homologs of ORF68.

**Figure supplement 4.** ORF68 remains oligomeric in the presence of high monovalent salt.

**Figure supplement 5.** Extension of the C-terminal tail of ORF68 reduces protein levels and prevents infectious virion production.

van der Waals contacts and a stacking interaction between F59 of one subunit and F335 of the adjacent subunit. This interface is generally poorly conserved, with the exception of N150 and D331, which interact with the backbone of neighboring subunits. A loop consisting of residues 324–329 inserts into an adjacent monomer to contact residues 136–145 and 55–59, an interaction that appears to be stabilized by the first zinc finger (*Figure 1c*).

The C-terminal tail of each ORF68 monomer is buried near the interface with its neighbor. The penultimate residue, Y466, is perfectly conserved in all homologs and located in the pentamer structure near the highly conserved residues H364 and C369 (*Figure 1—figure supplement 3*, *Figure 1—figure supplement 5a*). Furthermore, the C-terminal carboxyl group is surrounded by the highly conserved residues K365, D370, K426, and Q429 (*Figure 1—figure supplement 5b*). We found that addition of a C-terminal tag reduces protein levels and prevents infectious virion production (*Figure 1—figure supplement 5c–g*). This effect was also observed in HCMV, where C-terminal tagging of the ORF68 homolog, UL52, prevented virion production and led to an aberrantly disperse localization throughout the cell (*Borst et al., 2008*). Thus, disrupting the coordination of ORF68's C-terminus at the subunit interface through addition of a tag likely interferes with pentamer formation, destabilizes the protein, and prevents its function in DNA packaging and cleavage.

## Zinc fingers in ORF68 and its homologs are necessary for stability

ORF68 contains several motifs with highly conserved cysteines and histidines. These residues form three CCCH-type zinc fingers, as supported by the tetrahedral coordination geometry, a large anomalous scattering signal, and C/H composition of the putative zinc fingers, along with the previous observation that the homolog from HSV-1, UL32, binds zinc (*Chang et al., 1996*). The residues that compose two of these zinc fingers (residues C52, C55, H130, and C136 in the first zinc finger; residues C296, C299, H366, and C373 in the second zinc finger) are perfectly conserved in all identified homologs of ORF68 (*Figure 2a, b*, *Figure 1—figure supplement 3*). The third zinc finger (consisting of C191, C192, C415, and H452) is generally conserved in the alpha- and gammaherpesviruses (with the exception of the closely related model murine gammaherpesvirus MHV68), but is missing in the beta-herpesviruses (*Figure 2c*, *Figure 1—figure supplement 3*). This third zinc finger is also atypical in that two coordinating residues, C191 and C192, are adjacent to each other. Such noncanonical cysteine organization in a zinc finger is rare, but has been observed in the structure of RPB10, a subunit of RNA polymerase (*Mackereth et al., 2000*). There is additional weak anomalous density for a potential fourth metal binding site, coordinated by C169, C172, and H464, yet this was not included in the final model due to inconsistent density and stereochemistry.

To evaluate the role of individual residues in Zn coordination, we generated ORF68 mutants containing cysteine to alanine substitutions within the zinc fingers or for a similarly conserved cysteine outside of the zinc finger motifs (residue C79) (*Figure 2d*). ORF68 variants with a mutation in any of the zinc fingers were poorly expressed in transfected HEK293T cells, suggesting that the zinc fingers are required for structural stability of the protein. Several conserved cysteines in UL32 were previously identified as important for virion production in HSV-1 (*Albright et al., 2015*). We observed that these cysteines in UL32 are homologous to the zinc finger residues in ORF68 and are similarly essential for structural stability as their substitution with alanine resulted in lower UL32 protein levels, whereas mutation of a conserved cysteine outside of the zinc fingers had no effect (*Figure 2e*). Thus, UL32 likely also contains zinc fingers that are required for its structural stability.

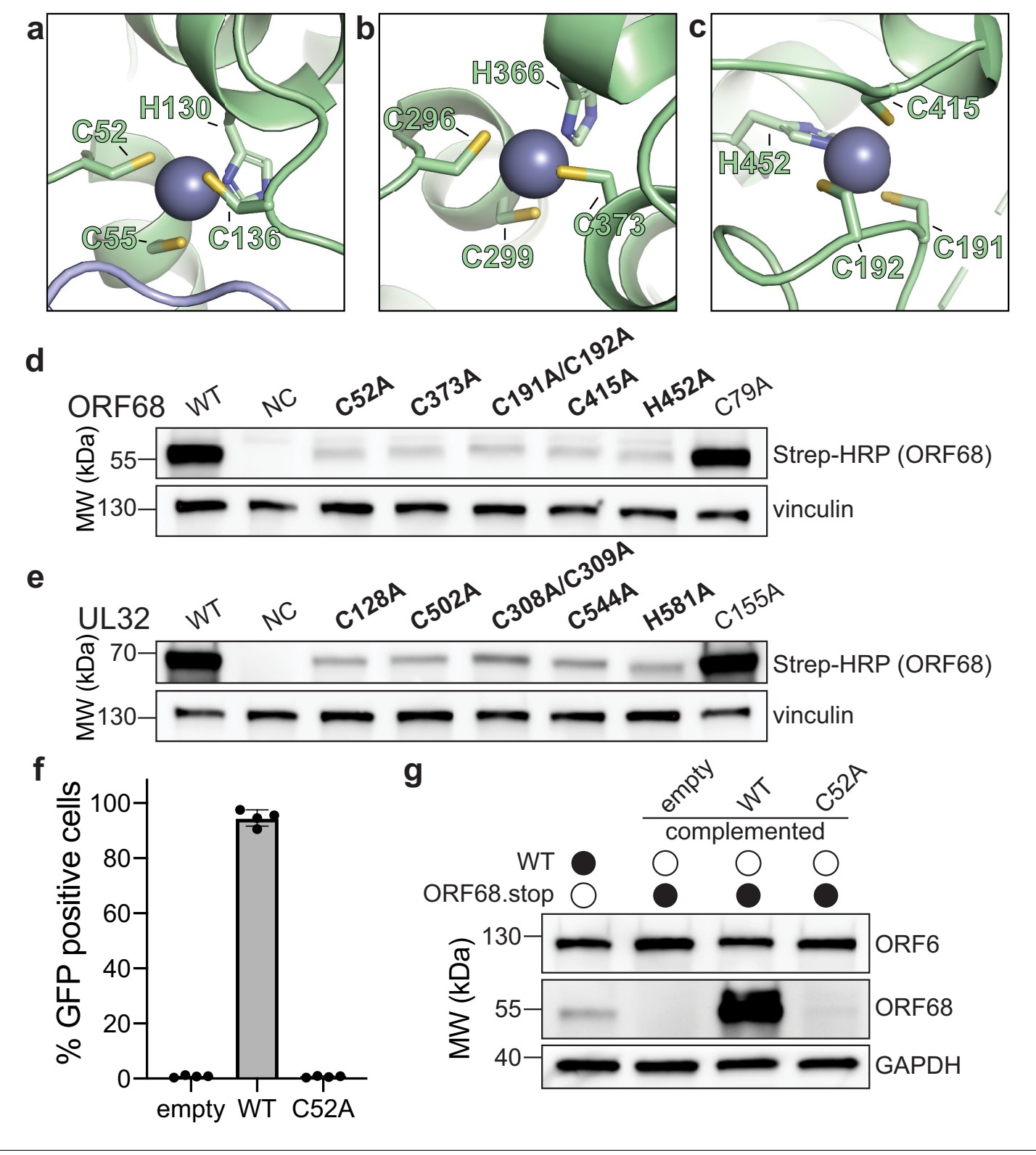

**Figure 2.** ORF68 and homologs are zinc finger-containing proteins. (a–c) The Zn$^{2+}$ ion within the three zinc finger motifs is shown as a blue-gray sphere, while coordinating cysteines and histidines are shown in sticks. (d, e) Western blot of whole cell lysate (33 μg) from HEK293T cells that were transfected with plasmids encoding wild-type or mutant variants of ORF68 (e) or UL32 (f). Vinculin serves as a loading control. (f) ORF68.stop iSLK cells

*Figure 2 continued on next page*

*Figure 2 continued*

were lentivirally transcomplemented with empty vector or with plasmids encoding wild-type or C52A ORF68. Progeny virion production by these cell lines was assayed by supernatant transfer and flow cytometry of target cells. (g) Western blot of transcomplemented ORF68.stop iSLK cells used in (e).

We next tested whether these stabilizing zinc fingers in ORF68 were required for the production of infectious virions. Using a latently infected inducible SLK (iSLK) BAC16 cell line in which the KSHV genome contains two premature stop codons to prevent ORF68 expression (*Gardner and Glaunsinger, 2018*), we assessed whether complementation with constitutively expressed wild-type ORF68 or the C52A zinc finger mutant allowed for production of infectious virions in a supernatant transfer assay. Wild-type ORF68-expressing cells were able to produce virions sufficient to infect nearly 100% of target cells, while cells expressing ORF68-C52A were unable to produce progeny virions (*Figure 2f*). ORF68-C52A could not be detected by western blot, suggesting that the C52A mutation is severely disruptive to the structure of ORF68 even in the context of viral infection (*Figure 2g*).

## Homologs of ORF68 form similar structures

ORF68 homologs can be found in all known members of the *Herpesviridae*, although BLAST cannot identify candidates in the *Alloherpesviridae* and *Malacoherpesviridae*, distantly related families in the order *Herpesvirales* that infect fish, amphibians, and mollusks. Homologs of ORF68 (467 residues) in the *Herpesviridae* range in size from 437 residues (MHV68 mu68) to 668 residues (HCMV UL52) and show generally low sequence conservation. Conserved residues in ORF68 are generally located in the hydrophobic core of ORF68, while variable residues map to the surface, particularly the outer faces of the ring (*Figure 3—figure supplement 1*). Most of the differences in length across the homologs come from N-terminal extensions or insertions in surface-exposed loops, suggesting a conserved core structure that is shared across the herpesviruses (*Figure 1—figure supplement 3*).

We sought to determine whether ORF68 homologs have a conserved structure, which may help identify features important for their function in packaging. We purified BFLF1, the ORF68 homolog in EBV, from transiently transfected HEK293T cells. Interestingly, size exclusion chromatography and negative stain EM revealed that BFLF1 forms decameric rings, consisting of two stacked pentamer rings (*Figure 3—figure supplement 2a, b*). We determined the structure of BFLF1 by cryo-EM at 3.60 Å resolution and found that it forms pentameric rings that are comparable in size to those of ORF68 and with a highly similar structure (*Figure 3—figure supplement 2c*). The BFLF1 structure is highly similar to that of ORF68, with a Root-mean-square deviation (RMSD) of 1.09 for the monomers and an overall RMSD of 1.95 for the pentameric complexes (*Figure 3a*). The internal core of the protein, including the first two zinc fingers (residues C54/C57/H132/C138 and C316/C319/H386/C393 in BFLF1), is structurally highly conserved (*Figure 3b*). Small variations occur in surface-exposed loops. Residues 180–240 (corresponding to residues 178–219 in ORF68) and 264–278 (corresponding to residues 244–258 in ORF68) could not be resolved, suggesting inherent flexibility. This flexibility also prevented unambiguous modeling of the putative third zinc finger, although BFLF1 residues C432 and H469 can be modeled, and C214/C215 likely lie within the immediately surrounding disordered region.

Based on the high structural similarity between ORF68 and BFLF1, we wondered if homologs from more distantly related herpesviruses retain a similar fold. Despite overall low sequence conservation across ORF68 homologs in the *Herpesviridae*, 24 residues are perfectly conserved, the majority of which are in the hydrophobic core of the protein, including the 8 residues involved in zinc fingers (*Figure 1—figure supplement 3*). Several of the conserved residues, including N150 and D331, sit at the subunit interface. Only one sidechain is surface-exposed (N325). Additionally, large insertions or deletions relative to ORF68 occur at surface-exposed loops or unstructured regions (*Figure 1—figure supplement 3*). Thus, despite overall low sequence identity, the structural core of various homologs is likely similar to ORF68, with differences arising in surface-exposed loops of variable length. We performed negative stain EM on the homolog from HSV-1, UL32, to determine if it also retains homopentameric quaternary structure (*Figure 3c*). Despite its low overall sequence homology to ORF68 and BFLF1, UL32 formed ring-shaped structures as well, and class averages

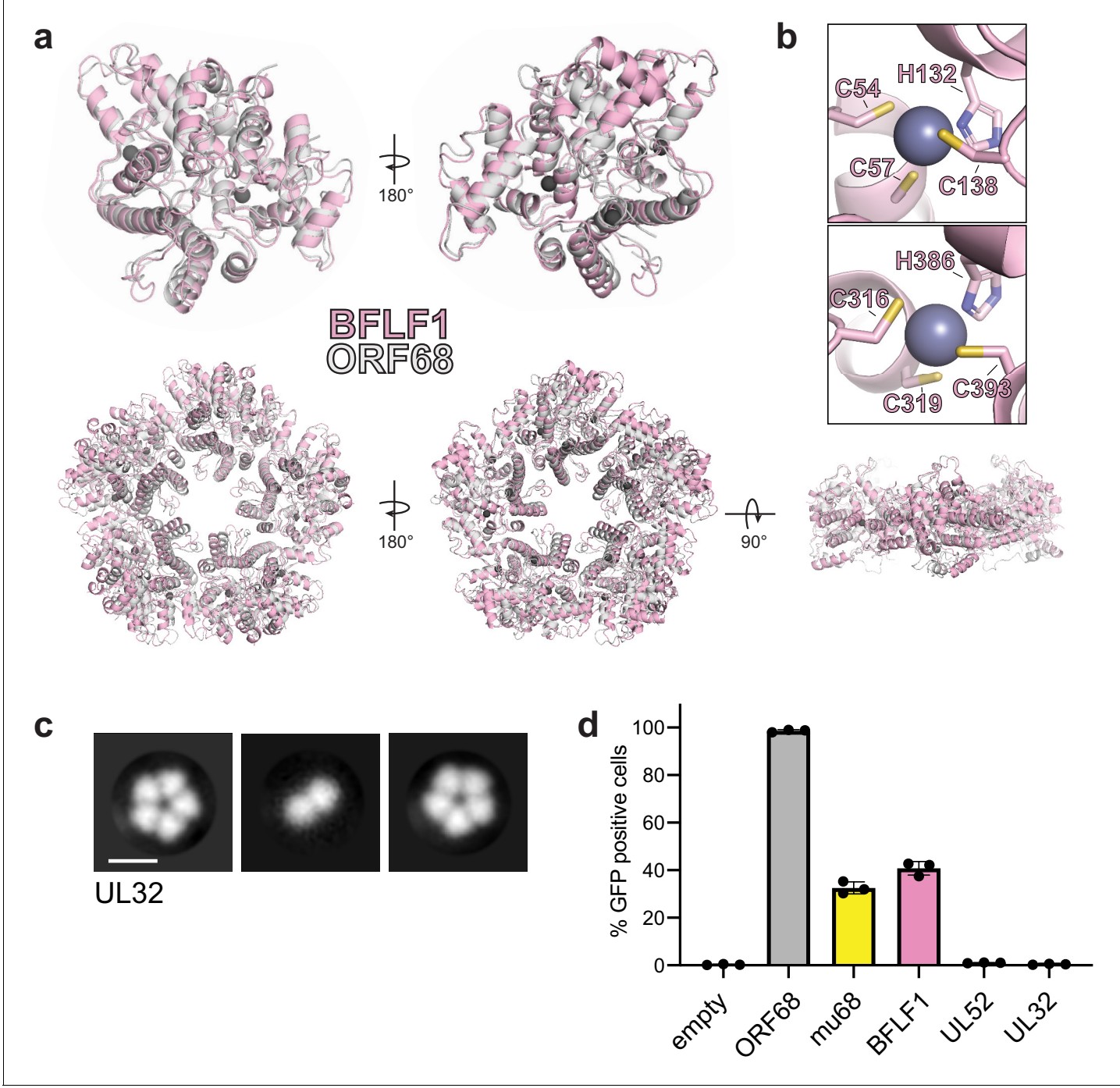

**Figure 3.** Homologs of ORF68 possess similar structures. (a) Overlay of ORF68 (gray) and BFLF1 (pink) monomers (top) and their homopentameric complexes (bottom). (b) BFLF1 contains at least two zinc fingers, with $Zn^{2+}$ ions shown as a blue-gray sphere. (c) Representative 2D class averages from negative stain EM of UL32. Scale bar = 100 Å. (d) ORF68.stop iSLK cells were lentivirally transcomplemented with plasmids encoding N-terminally Strep-tagged ORF68 or homologs from EBV (BFLF1), MHV68 (mu68), HCMV (UL52), or untagged HSV-1 (UL32). Progeny virion production by these cell lines was assayed by supernatant transfer and flow cytometry of target cells.

The online version of this article includes the following figure supplement(s) for figure 3:

**Figure supplement 1.** ConSurf model of ORF68.

**Figure supplement 2.** EM data analysis for BFLF1.

suggested that it forms a homopentamer. The ability to form a pentameric ring is therefore a common property of ORF68 and its homologs.

Next, we investigated if homologs were sufficiently similar to functionally replace ORF68's essential role in viral packaging in KSHV. We generated stable cell line derivatives that constitutively expressed either ORF68 or its homolog from EBV (BFLF1), MHV68 (mu68), HCMV (UL52), or HSV-1 (UL32) to complement an ORF68-null virus (*Figure 3d*). As expected, ORF68.stop cells complemented with ORF68 allowed for production of virions that infected nearly 100% of target cells, while complementation with empty vector failed to produce infectious virions. Homologs from the gamma-herpesviruses, BFLF1 and mu68, were able to partially complement loss of ORF68, although not to levels comparable to ORF68-expressing cells. In contrast, homologs from the more distantly related alpha- and beta-herpesviruses, UL32 and UL52, failed to complement deletion of ORF68. BFLF1 and mu68 are most similar to ORF68 (35% and 33% identity, respectively), while UL32 and UL52 are more diverged in sequence (25% and 22% identity, respectively). Thus, although homologs across the herpesviruses share a common core fold and homopentameric architecture, other features or the identity of surface residues likely play a role in their function and in the interaction with other components during DNA packaging.

## ORF68 binds dsDNA via its positively charged central channel

The calculated electrostatic surface of ORF68 shows a striking colocalization of positive charges on one side of the ring around the entrance of the central channel (*Figure 4a*). We previously observed that ORF68 can bind an 800 bp dsDNA probe corresponding to the GC-rich terminal repeat of KSHV (*Gardner and Glaunsinger, 2018*), and therefore hypothesized that the electrostatic surface around the central channel of ORF68 could be important for dsDNA binding. Although we also previously observed that ORF68 has weak nuclease activity (*Gardner and Glaunsinger, 2018*), we were unable to identify a motif consistent with such activity in the structure. ORF68 binds 10–20 bp dsDNA with high affinity, while multiple binding events are observed on longer probes (*Figure 4—figure supplement 1a*; *Gardner and Glaunsinger, 2018*). We selected a series of surface-exposed positively charged residues (arginine or lysine) on either side and throughout the central channel to test if substitution to alanine reduced dsDNA binding. All ORF68 mutants expressed and purified similar to wild-type ORF68 (*Figure 4—figure supplement 2a*), and a representative mutant (ORF68 K435A) behaved similar to wild-type protein in size exclusion chromatography, suggesting that mutations in the central channel do not affect the homopentameric architecture (*Figure 4—figure supplement 2b*). We used electrophoretic mobility shift assays to test the relative dsDNA-binding affinity of mutants compared to wild-type ORF68 (*Figure 4b, c*, *Figure 4—figure supplement 2c*). Mutations K435A (top, more constricted side of the central channel), R443A (middle of the central channel), and K174A/R179A/K182A (henceforth referred to as '3+', in the disordered region of the central channel) all resulted in a drastic reduction in binding affinity, with negligible interactions even at 4 μM ORF68. The K450A/R451A mutation (bottom of the central channel) bound with lower affinity than wild-type ORF68, as indicated by the lack of concrete bands and a consequential smear. In contrast, mutation in two sets of surface-exposed positive residues located outside the channel, R14A/K310A and K395A/K396A, bound dsDNA comparably to wild-type ORF68 with an effective binding affinity (EC50) of ~25 nM. Thus, while charge mutations on the periphery of the ring have no effect, mutations within the channel – and specifically near the more constricted top portion – are deleterious to dsDNA binding, likely because residues from all subunits form a large binding interface with high charge density in and near the central channel.

Herpesviral DNA packaging requires both sequence-specific and nonspecific DNA binding. Site-specific binding and cleavage within the terminal repeats is required to ensure full-length genomes are packaged; however, this role is thought to be fulfilled by the terminase (*Adelman et al., 2001*; *Theiß et al., 2019*). Packaging also depends on nonsequence-specific interactions with various factors, such as HSV-1 UL25 (KSHV ORF19) that binds DNA and has recently been shown to be the 'portal cap' that prevents DNA escape after packaging has been completed (*Gong et al., 2019*; *Liu et al., 2019*; *Ogasawara et al., 2001*). To assess the sequence specificity of ORF68, we compared its binding to a 20 bp sequence derived from the terminal repeats with high (85%) GC content versus a scrambled sequence with 50% GC content (*Figure 4—figure supplement 1b*). ORF68 bound with ~25 nM affinity to both substrates, suggesting that it recognizes dsDNA in vitro in a nonsequence-specific manner.

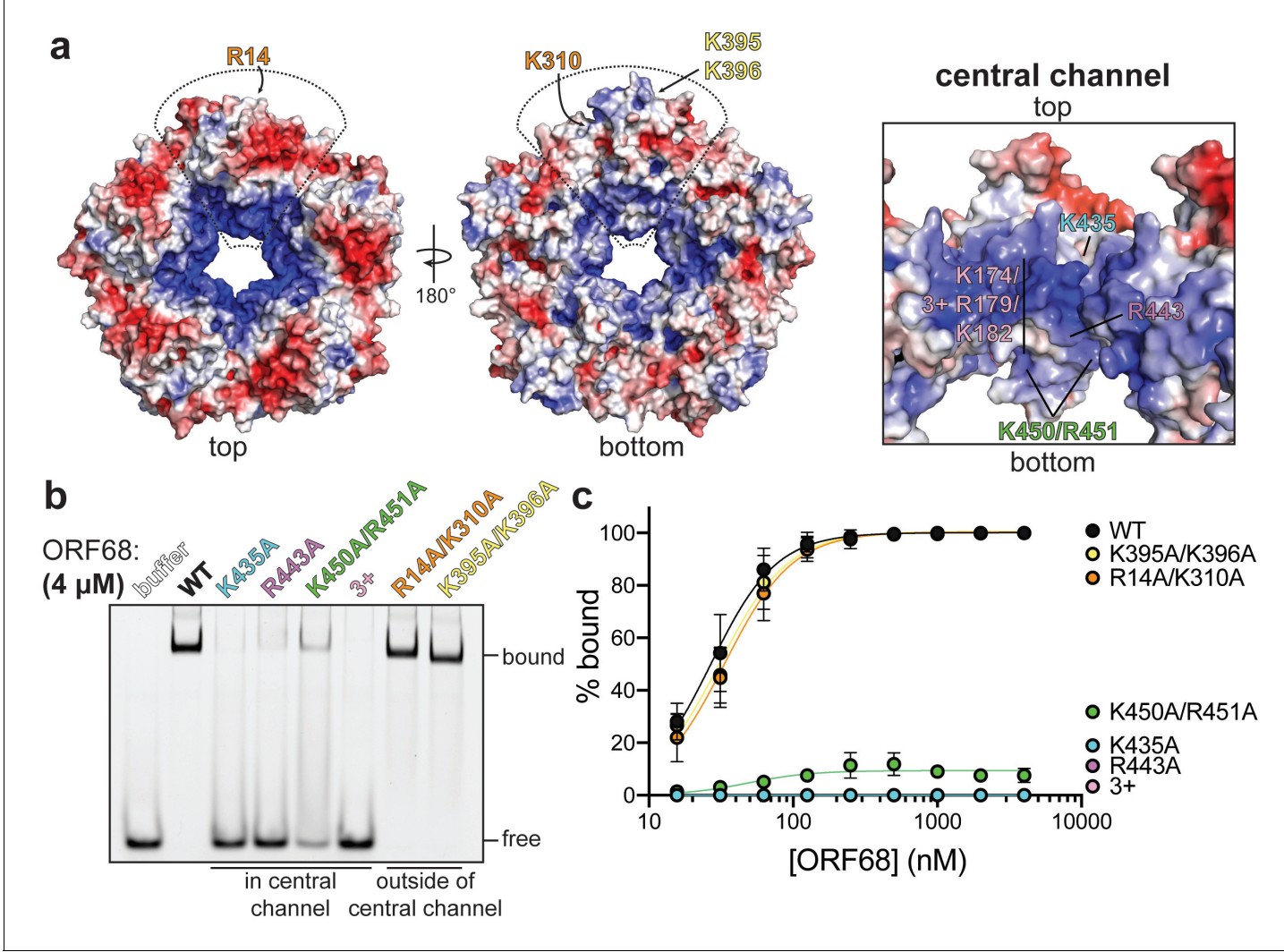

**Figure 4.** ORF68 binds nucleic acid in vitro via its central channel. (a) Electrostatic surface of the ORF68 pentamer, contoured from +5 kT/e (blue) to −5 kT/e (red) and shown from the top (left), bottom (middle), and through the central channel (right). The electrostatic surface lacks regions that were disordered in the structure, including residues 169–172 and 179–188, which face the central channel. The locations of residues selected for mutation are indicated on one monomer of the pentamer. (b) Electrophoretic mobility shift assay using fluorescein-labeled 20 bp dsDNA probe (10 nM) and wild-type or mutant ORF68 (4 µM). (c) Binding curves for wild-type and mutant ORF68 interacting with the 20 bp dsDNA probe were determined by electrophoretic mobility shift assays as in (b). Data represent the mean ± s.d. of three independent experiments. Data were fit with a nonlinear regression to the Hill equation.

The online version of this article includes the following figure supplement(s) for figure 4:

**Figure supplement 1.** ORF68 nonspecifically binds nucleic acid.

**Figure supplement 2.** ORF68 mutants can be purified, but mutations in the central channel prevent dsDNA binding.

## Positively charged residues within the central channel are required for cleavage and packaging

We next sought to determine whether electrostatically mediated nucleic acid binding is important during viral replication. Using the Red recombinase system (*Brulois et al., 2012*), we incorporated two ORF68 variants that ablate dsDNA binding in vitro, K435A and the 3+ mutant, as well as the corresponding mutant rescue (MR) control constructs with revertant mutations into the KSHV BAC16 genome (*Figure 5—figure supplement 1a, b*). We established latently infected iSLK cell lines harboring KSHV with a mutant copy of ORF68 and assessed their ability to produce infectious virions using the supernatant transfer assay (*Figure 5a*). As expected, wild-type KSHV was able to infect nearly 100% of target cells, whereas the ORF68.stop cells failed to produce infectious virions. The

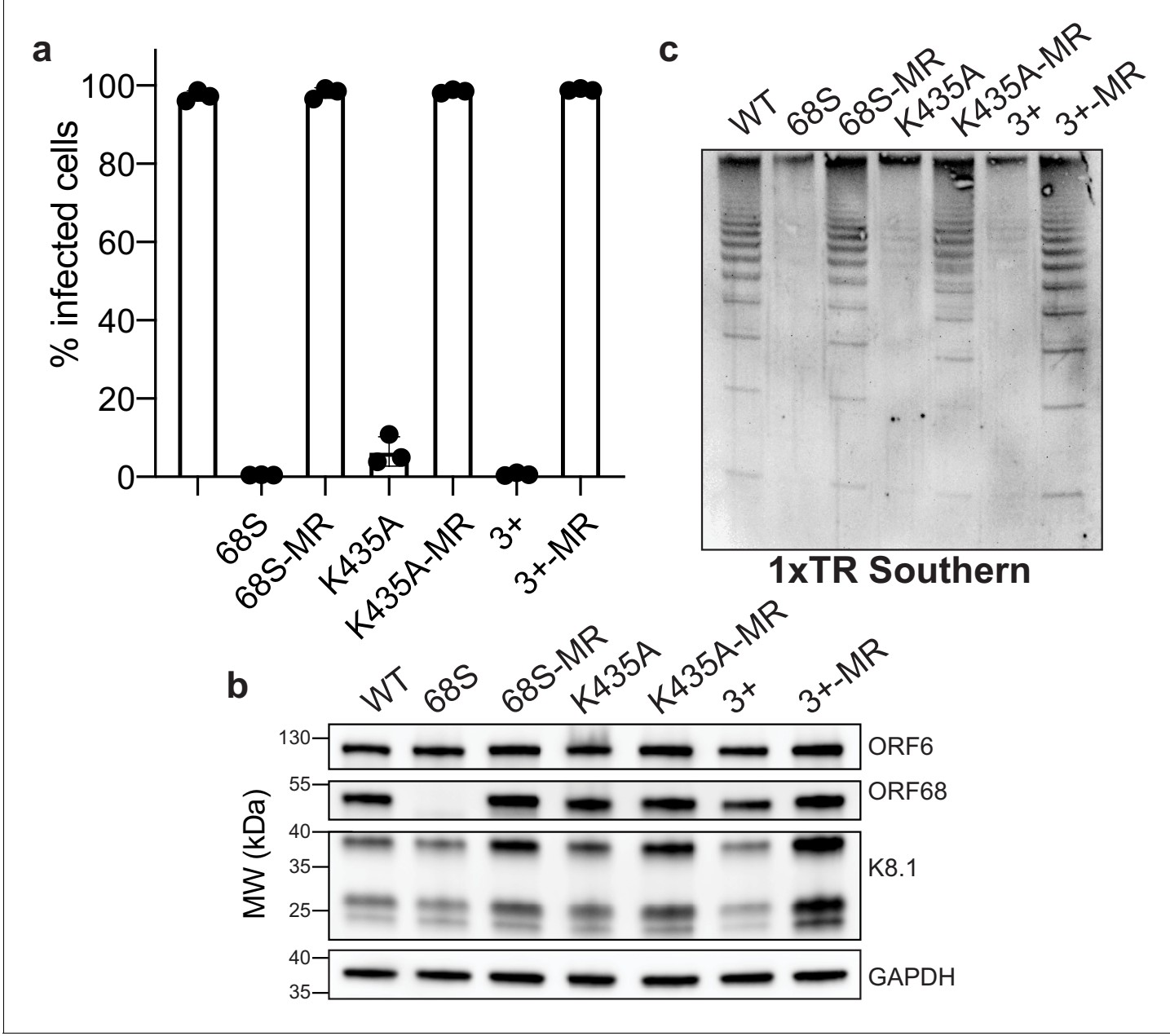

**Figure 5.** Residues in ORF68 that ablate dsDNA binding in vitro are required for genome cleavage and packaging in vivo. (a) iSLK cell lines containing ORF68 mutants (68S, K435A, and 3+) and their corresponding mutant rescues (MR) were established using the KSHV BAC16 system. Progeny virion production by these cell lines was assayed by supernatant transfer and flow cytometry of target cells. (b) Western blot of whole cell lysate (25 µg) from ORF68.stop iSLK cell lines. GAPDH was used as a loading control. ORF6 is an early gene and K8.1 is a late gene. (c) Southern blot of DNA isolated from iSLK cell lines using a probe for the terminal repeats. DNA was digested with PstI, which cuts within the genome but not within the terminal repeats and generates a ladder of terminal repeat-containing DNA when successful cleavage and packaging occurs.

The online version of this article includes the following figure supplement(s) for figure 5:

**Figure supplement 1.** Construction and validation of mutant viruses.

ORF68-K435A cells had a pronounced virion-production defect, leading to infection of <5% of target cells, and the ORF68-3+ cell line failed to produce any infectious virions. Both the ORF68-K435A and 3+ MR cell lines behaved similar to wild-type KSHV-infected cells, confirming that defects in virion production are caused by the charge mutations in ORF68, rather than effects on neighboring genes or mutations elsewhere in the viral genome acquired during recombination. Importantly, the

K435A and 3+ mutations retain wild-type levels of ORF68 expression (*Figure 5b*). Interestingly, a small but consistent reduction in the expression level of the late gene K8.1 can be observed in the ORF68.stop, ORF68-K435A, and ORF68-3+ cell lines, suggesting that ORF68 may have other minor roles in gene expression or protein homeostasis during infection.

Given the close association of packaging with replication, we tested if the charge mutations in ORF68 have an effect on DNA replication. We measured viral DNA replication by qPCR and found no difference between the wild-type virus and the ORF68.stop, ORF68-K435A, or ORF68-3+ viruses (*Figure 5—figure supplement 1c*). While the K435A MR virus had levels comparable to both wild-type and the K435A mutant, the rescue cell lines for the ORF68 null and 3+ mutants had higher levels of replication, which may be due to changes in the efficiency of cell line establishment.

Given the drastic defects observed in virion production for the K435A and 3+ mutants, we sought to determine if these mutations act at the step of cleavage and packaging. Using an assay that relies on the intimate coupling between genome packaging and cleavage that probes the cleavage state of terminal repeat by Southern blot, we previously demonstrated that an ORF68-null virus is defective for viral genome packaging (*Gardner and Glaunsinger, 2018*). Similar to cells lacking ORF68, the ORF68-K435A and 3+ cell lines reveal prominent defects in genome cleavage that are rescued by their respective MR lines (*Figure 5c*). Thus, the virion production defect observed in the K435A and 3+ cell lines is caused by a failure to properly cleave and package the viral genome, suggesting that the ability of ORF68 to bind nucleic acid through its positively charged central channel is required for successful packaging.

## Discussion

Here, we present the structure of the only functionally undefined component of the herpesviral packaging machinery and reveal that its ability to bind DNA, likely involving its positively charged central channel, is critical for genome packaging. KSHV ORF68 adopts a novel homopentameric ring structure with largely α-helical monomers stabilized by multiple zinc fingers. Comparison to the EBV BFLF1 structure reveals extremely high structural homology, and negative stain EM analyses of a homolog from a further diverged alphaherpesvirus suggest a common monomer fold and potential to form pentameric quaternary structure. Thus, it is likely that the core architecture is conserved across herpesviruses from all subfamilies. These structures provide an important framework for mechanistic dissection of the roles that ORF68 and its homologs play in herpesviral packaging.

The importance of ORF68 and its homologs for genome cleavage and packaging has been studied in several herpesviruses, and deletion of these proteins results in a common phenotype, indicative of a conserved function (*Albright et al., 2015*; *Gardner and Glaunsinger, 2018*; *Lamberti and Weller, 1998*; *Borst et al., 2008*). However, our understanding of the packaging process lacks a clear role for this protein (*Heming et al., 2017*; *Figure 6a*). UL32, the homolog from HSV-1, has been described as a glycoprotein (*Becker et al., 1988*), yet functional and structural analyses by us and others suggest that this is not the case (*Chang et al., 1996*). Furthermore, despite earlier work suggesting that UL32 was involved in localizing capsids to replication compartments during packaging, corresponding defects in localization were not observed for a virus containing a full deletion of UL32 (*Albright et al., 2015*; *Lamberti and Weller, 1998*). More recent work proposed that UL32 is important for regulation of disulfide bond formation during infection (*Albright et al., 2015*), which was based on the observation that several cysteine-rich motifs are required for infectious virion production. This is consistent with our finding that these cysteines in ORF68 and UL32 are involved in zinc finger motifs and thus play critical roles for stability. Studies in HSV-1 and HCMV demonstrate that the homologs of ORF68 are not involved in expression or localization of the portal or terminase as deletion of UL32 or UL52, respectively, has no effect on these properties (*Borst et al., 2008*; *Yu and Weller, 1998*). A plethora of mass spectrometry data from different purified herpesviruses, along with recent cryo-EM reconstructions of herpesvirus capsids, suggest that neither ORF68 nor its homologs are stably capsid-associated or packaged into virions. Thus, although it is well established that ORF68 and its homologs are essential for packaging, their role in this process has remained elusive. Given the properties of ORF68 that we have identified – namely, conserved pentameric symmetry with a positively charged channel important for dsDNA binding and virion production – what role could ORF68 be playing during packaging?

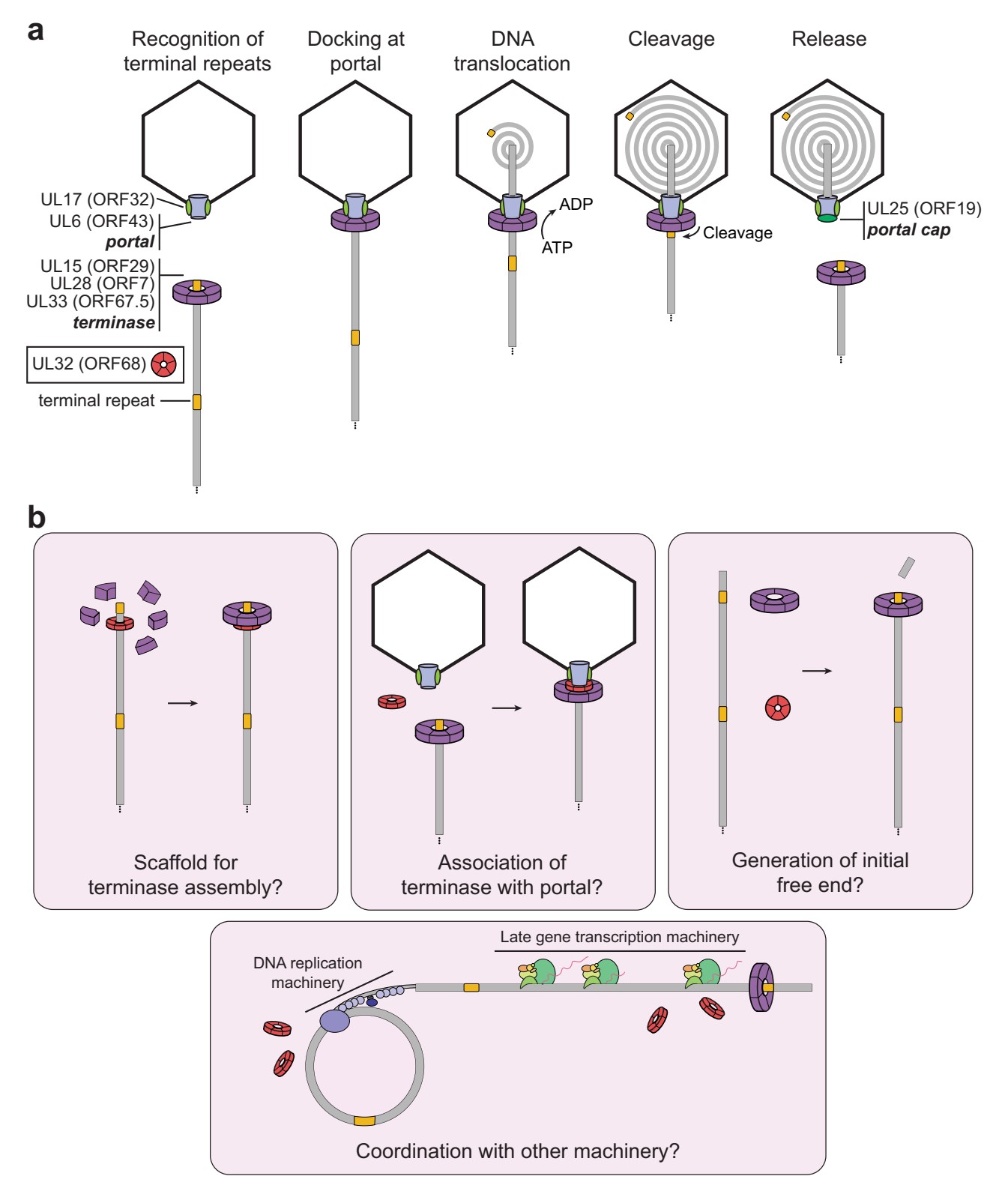

**Figure 6.** Model of herpesviral cleavage and packaging. (a) Genes required for cleavage and packaging in HSV-1 and their homologs in KSHV (listed in brackets) are listed. UL15, UL28, and UL33 form the terminase complex that must dock with the portal protein (UL6), capsid, and capsid-associated proteins (including UL17). The terminase translocates the dsDNA genome into the capsid and cleaves within the terminal repeats once a full unit-length genome has been packaged. After release of the remaining genome, the UL25 portal cap binds to stabilize the packaged genome. The precise role of

*Figure 6 continued on next page*

Figure 6 continued

UL32 (KSHV ORF68) has not been determined. (**b**) Possible roles of ORF68 during packaging could include acting as a scaffold for the terminase to bind the nascent genome (left), acting as an adaptor terminase association with the portal (middle), or promoting formation of the initial free end on nascent genomes (right). Further potential roles include interfacing with the DNA replication machinery or late gene transcription machinery (bottom).

The online version of this article includes the following figure supplement(s) for figure 6:

**Figure supplement 1.** Model of DNA binding in the central channel of ORF68.

Similarities between the capsid structure, portal, and terminase in tailed bacteriophages and herpesviruses strongly suggest that these viruses share ancestrally ancient packaging machinery (*McGeoch et al., 2006*). Although ORF68 has no identifiable structural or functional homolog in phages, several previously studied model phages have accessory factors critical for efficient packaging. These factors highlight possible analogous roles of ORF68 and its homologs in herpesvirus packaging. In phage λ, the factor gpFI is required for efficient packaging (*Becker et al., 1988*; *Davidson and Gold, 1987*; *Sippy and Feiss, 2004*). gpFI is thought to facilitate binding of the DNA-engaged terminase to the capsid (prohead) through interactions with the major head protein (*Becker et al., 1988*; *Davidson and Gold, 1987*; *Popovic et al., 2012*). Phage phi29 encodes a pentameric structural RNA (pRNA) that is sandwiched between the portal and terminase during packaging (*Ding et al., 2011*; *Simpson et al., 2000*; *Woodson et al., 2020*). Although structurally unrelated to gpFI or pRNA, ORF68 by analogy may act as a bridge between the portal and the terminase. Recent reconstructions of the HSV-1 and KSHV capsids (*Gong et al., 2019*; *Liu et al., 2019*) revealed their structures at atomic resolution and resolved the portal vertex through which the DNA genome is packaged. The portal cap, which has pentameric symmetry, sits atop this vertex and is thought to be composed of HSV-1 UL25 (KSHV ORF19). Notably, binding of the portal cap to the portal vertex reveals how proteins with pentameric symmetry, like ORF68, could interface with the dodecameric portal.

We propose that ORF68's role in packaging is to assist docking of the DNA-bound terminase complex with the portal machinery, perhaps by acting as a scaffold that confers fivefold symmetry for the terminase (*Figure 6b*). Nonspecific DNA binding by ORF68 could promote formation of the terminase–DNA complex and drive the packaging reaction forward. The genome may thread through the channel of the pentameric ring, which in its most constricted region is just wide enough to accommodate dsDNA (*Figure 6—figure supplement 1*). Although a recent structure of the terminase complex from HSV-1 revealed hexameric symmetry in solution, its symmetry while bound to the portal remains unknown (*Yang et al., 2020*). It is well established that phage motors can adopt different stoichiometries, but actively packaging motors are thought to be pentameric (*Woodson et al., 2020*; *Hilbert et al., 2015*; *Sun et al., 2008*). Future work should seek to determine whether ORF68 threads nucleic acid through its central channel, whether it makes stable or transient protein–protein interactions with components of the packaging machinery, and whether it influences the assembly or stoichiometry of the terminase complex.

Alternatively, ORF68 could be involved in the initial generation of a free dsDNA end for packaging, a process which is poorly understood. Herpesvirus genome replication produces branched, head-to-tail concatemers that are cleaved within a repeated region (the terminal repeats in KSHV [*Lagunoff and Ganem, 1997*]; the directly repeated element [DR1] within the *a* sequence at the $U_L$ terminus in HSV-1) during packaging to produce a packaged capsid containing a unit-length genome (reviewed in *Heming et al., 2017*). Since replication of the viral genome does not initiate within these repeated regions, an initial cleavage event to produce a substrate for packaging must occur. Of the seven herpesviral genes required for successful packaging (HSV-1 UL6, UL15, UL17, UL25, UL28, UL32, and UL33), when deleted, all of these except UL25 have the additional effect of preventing cleavage, which is closely tied to packaging. ORF68 and its homologs may thus be involved in either the preparation of the initial substrate for packaging (generation of the initial cut) (*Figure 6b*) or regulation of the cleavage event during subsequent rounds of packaging. An additional possibility could include a role for ORF68 and its homologs in the coordination of packaging with DNA replication and late gene machinery, as coupling between these processes has been observed in phage (*Black and Peng, 2006*), and several capsid components have been shown to interact with replicating viral genomes in HSV-1 (*Dembowski et al., 2017*).

Although ORF68 and its homologs share a conserved structure and likely play similar roles in packaging across the herpesviruses, they are expressed with divergent kinetics, suggesting potential additional functions. HSV-1 UL32 and HCMV UL52 are late genes, consistent with a primary role in packaging (*Albright et al., 2015*; *Borst et al., 2008*), whereas KSHV ORF68 is an early gene, accumulating prior to the expression of several capsid proteins (*Gardner and Glaunsinger, 2018*). It remains to be determined whether ORF68 has additional roles early in infection. We demonstrated that deletion of ORF68 can be partially complemented by other gammaherpesvirus homologs (mu68 and BFLF1). Interestingly, deletion of the ORF68 homolog in the alphaherpesvirus pseudorabies virus (PrV) can be partially complemented by expression of HSV-1 UL32; however, the inverse complementation was not observed, indicating that some functions or interactions are sufficiently similar for PrV virion production but not HSV-1 virion production (*Fuchs et al., 2009*). PrV UL32 and HSV-1 UL32 share ~52% sequence similarity, as do ORF68 and EBV BFLF1 or MHV mu68, suggesting that in both cases common residues or structural motifs mediate shared interactions to carry out critical functions in the cell. Future high-resolution structural studies of other homologs may identify shared and divergent features, and it will be interesting to investigate how homologs lacking the third zinc finger motif (i.e., HCMV UL52) are structurally stabilized. While we found that ORF68, BFLF1, and UL32 are all capable of forming pentameric rings, the significance of further oligomerization (e.g., the stacked pentamers of BFLF1) remains to be determined. Future work should also seek to determine if the oligomerization state of ORF68 and its homologs influences assembly of the packaging machinery and what oligomeric state is adopted in the context of viral infection.

Despite widespread prevalence of herpesviruses and the significant diseases they cause, there is no cure for any herpesvirus, and a vaccine exists only for the alpha-herpesvirus varicella zoster virus. The majority of antiherpesviral drugs target the DNA replication machinery, but have several disadvantages, including the emergence of resistance mutations and a narrow spectrum of use (*Coen and Schaffer, 2003*; *Poole and James, 2018*; *Weller and Kuchta, 2013*). However, Letermovir, a drug recently approved by the US Food and Drug Administration for the treatment of HCMV in stem cell transplant recipients, targets the HCMV terminase through an unknown mechanism (*Goldner et al., 2011*; *Ligat et al., 2018*; *Lischka et al., 2010*). Letermovir lacks activity against other herpesviruses (*Marschall et al., 2012*), suggesting that key differences exist in their conserved machinery. Our work represents a major advance in understanding the complexities of herpesviral DNA packaging and the mechanistic role of one of its essential components. Additional work will be required to elucidate the detailed role of ORF68 and its homologs in packaging. Understanding the molecular underpinnings of packaging and the differences in mechanism across the alpha-, beta-, and gamma-herpesviruses is critical for the future development of antiherpesvirals as effective therapeutics.

## Materials and methods

### Plasmids

ORF68 was amplified from pcDNA4/TO-ORF68-2xStrep (*Gardner and Glaunsinger, 2018*), and mu68 was amplified from MHV68 BAC DNA (*Adler et al., 2000*). BFLF1 cDNA was generated from EBV-infected Akata cells reactivated with anti-human IgG (*Takada and Ono, 1989*). UL52 was amplified from HCMV Towne DNA. UL32 was amplified using nested PCR from HSV-1 KOS DNA. ORF68 and its homologs were subcloned into the NotI and XhoI sites of pcDNA4/TO-2xStrep (N-terminal) using InFusion cloning (Clontech) (Addgene #162625–162629). C-terminally 2xStrep tagged ORF68 was previously described (*Davis et al., 2015*) (Addgene #136229). ORF68 was also subcloned into the EcoRV site of pcDNA4/TO-2xStrep (C-terminal) using InFusion cloning to generate pcDNA4/TO-ORF68 (tagless) (Addgene #166025). Mutations in ORF68 (Addgene #162630–162643) and UL32 (Addgene #162644–162649) were generated using inverse PCR site-directed mutagenesis with Phusion DNA polymerase (New England Biolabs) with primers as listed in *Supplementary file 3*. PCR products were DpnI-treated, ligated using T4 PNK and T4 DNA ligase, and transformed into *Escherichia coli* XL-1 Blue cells.

The expression plasmid for ORF68 was previously described (*Gardner and Glaunsinger, 2018*) (Addgene #162650) and is a pHEK293 UltraExpression I vector (pUE1-TSP) (Clontech) that encodes an N-terminal Twin-Strep tag and the coding region for ORF68 including its native start codon,

separated by an HRV 3C protease cleavage site. This plasmid was used as a template for inverse PCR to generate linearized pUE1-TSP vector containing the Twin-Strep tag and HRV 3C site. BFLF1, UL32, and mutants of ORF68 were subcloned from their respective pcDNA4/TO vectors into linearized pUE1-TSP vector using InFusion cloning to generate expression constructs (Addgene #162651–162658).

Plasmids for lentiviral transduction (pLJM1-2xStrep or untagged wild-type ORF68, mutants, and homologs) (Addgene #162659–162664) were generated by subcloning into the AgeI and EcoRI sites of pLJM1 modified to confer resistance to zeocin (Addgene #19319) using InFusion cloning. Lentiviral packaging plasmids psPAX2 (Addgene plasmid #12260) and pMD2.G (Addgene plasmid #12259) were gifts from Didier Trono. Details for plasmids used in this study can also be found in the Key Resources Table (Appendix).

## Transfections

HEK293T cells were plated and transfected after 24 hr at 70% confluency with PolyJet (SignaGen) or polyethylenimine (PEI). Cells were harvested after 24 hr (for expression studies) or 48 hr (for large-scale protein expression). For analysis of protein expression, cells were washed with phosphate buffered saline (PBS), pelleted at $1000 \times g$ for 5 min at 4°C, and lysed by resuspension in lysis buffer (150 mM NaCl, 50 mM Tris-HCl pH 7.4, 1 mM EDTA, 0.5% NP-40, and protease inhibitor [Roche]) with rotation at 4°C for 30 min. Lysates were clarified by centrifugation at $21,000 \times g$ for 10 min at 4°C. Lysate (20–33 µg) was used for SDS-PAGE and western blotting in Tris-buffered saline and 0.2% Tween 20 (TBST) using Strep-Tag II HRP (1:2500; EMD Millipore), rabbit anti-ORF68 (1:1000, *Gardner and Glaunsinger, 2018*), and rabbit anti-vinculin (1:1000; Abcam). Following incubation with primary antibodies, membranes were washed with TBST and imaged (for Strep-Tag II HRP) or incubated with goat anti-rabbit-HRP (1:5000; Southern Biotech).

## Protein expression and purification

Purification of Twin-Strep tagged ORF68 and BFLF1 for use in crystallography and cryo-EM was performed as described previously (*Gardner and Glaunsinger, 2018*). Briefly, pUE1-TSP-ORF68 or BFLF1 were transfected into ~70% confluent HEK293T cells using PEI. Cells were harvested after 48 hr and frozen at −80°C. Cells were lysed in lysis buffer (300 mM NaCl, 100 mM Tris-HCl pH 8.0, 5% glycerol, 1 mM dithiothreitol (DTT), 0.1% CHAPS, 1 µg/mL avidin, cOmplete, EDTA-free protease inhibitors [Roche]), rotated at 4°C for 30 min, sonicated to reduce viscosity, then centrifuged at $50,000 \times g$ for 30 min at 4°C. The lysate was filtered through a 0.45 µm filter, then purified using Strep-Tactin XT resin (IBA) in running buffer (300 mM NaCl, 100 mM Tris-HCl pH 8.0, 5% glycerol, 0.1% CHAPS, 1 mM DTT). Protein was eluted in running buffer containing 50 mM biotin, concentrated using a 30 kDa cutoff spin concentrator (Millipore), then the 2xStrep tag was removed by cleavage with HRV 3C protease (Millipore Sigma) overnight. Protein was further purified by size exclusion over a HiLoad 16/600 Superdex 200 pg column (GE Healthcare) in sizing buffer (100 mM NaCl, 20 mM HEPES pH 7.6, 5% glycerol, 1 mM Tris (2-carboxyethyl) phosphine hydrochloride (TCEP-HCl)). Size exclusion chromatography in *Figure 1—figure supplement 4*, *Figure 3—figure supplement 1b*, and *Figure 4—figure supplement 2b* was performed on a Superose 6 Increase 10/300 GL column (GE Healthcare) in sizing buffer or in sizing buffer containing concentrations of NaCl as indicated.

The native start methionine was included in the construct, and interestingly we reproducibly saw an ~50% distribution between translation initiation at the methionine before the Twin-Strep tag and at the native methionine for both ORF68 and BFLF1, but not UL32. Little untagged ORF68 or BFLF1 is lost during purification as one Twin-Strep tag in the pentamer is sufficient for enrichment on Strep-Tactin resin.

Proteins used for electrophoretic mobility shift assays and UL32 used in negative stain EM were purified as above, except that 0.5% CHAPS was used during lysis, 1 mM TCEP-HCl was used in lieu of DTT throughout the purification, the 2xStrep tag was not removed by incubation with HRV 3C protease (Millipore), and proteins were not purified by size exclusion chromatography.

## Negative stain and cryo-electron microscopy grid preparation, data collection, image processing, initial model building, and structure determination

For the preparation of negative-stain EM grids, ORF68, BFLF1, and UL32 were diluted to ~100–200 nM in dilution buffer (60 mM HEPES pH 7.6, 100 mM NaCl, 10 mM $MgCl_2$, 0.5 mM TCEP) and stained with 2% uranyl formate (pH 5.5–6.0) on thin carbon-layered 400 mesh copper grids (EMS) (*Ohi et al., 2004*). Micrographs were collected on a Tecnai 12 microscope (ThermoFisher) operated at 120 keV with 2.2 Å per pixel using a 4 k TemCam-F416 camera (TVIPS): 131, 79, and 100 total micrographs for ORF68, BFLF1, and UL32 datasets, respectively. Micrographs were CTF-corrected using CTFFIND4 (*Rohou and Grigorieff, 2015*) in Relion (*Scheres, 2012*). Single particles were automatically selected using Gautomatch (*Zhang, 2016a*): 57,510, 29,021, and 29,098 total particles for ORF68, BFLF1, and UL32 datasets, respectively, and 2D classification was performed in Relion (*Scheres, 2012*).

Cryo-EM grids were prepared by applying 3.5 µL of 5 µM (pentamer) ORF68 (in 50 mM HEPES pH 7.6, 50 mM NaCl, 50 mM KCl, 5% glycerol, and 1 mM TCEP) and 3.5 µL of 14 µM (pentamer) BFLF1 (in 60 mM HEPES pH 7.6, 100 mM NaCl, 50 mM KCl, 10 mM $MgCl_2$, 0.5 mM TCEP, 0.05% NP-40) to glow-discharged C-Flat holey carbon grids (CF-2/1–3 C-T, EMS). The samples were plunge-frozen using a Vitrobot (ThermoFisher) and imaged on a Talos Arctica TEM operated at 200 keV (ThermoFisher). Dose-fractionated imaging was performed by automated collection methods using SerialEM (*Mastronarde, 2005*). Data were collected as described in *Supplementary file 1*. Whole-frame drift correction was performed via Motioncor2 (*Zheng et al., 2017*) with dose weighting applied.

### ORF68 processing

Micrographs were CTF-corrected using GCTF (*Zhang, 2016b*) in Relion (*Scheres, 2012*). In total, 662,435 single particles were automatically selected using Gautomatch (*Zhang, 2016a*), from which 274,167 particles were selected from 2D class averages, generated in Relion. The 3D classification scheme is detailed in *Figure 1—figure supplement 1*. The final model was refined with C5 symmetry (pentamer) and sharpened using postprocessing to an estimated 3.37 Å. An alanine backbone was modeled manually in *Coot* (*Emsley et al., 2010*) to be used for phasing the crystal structure.

### BFLF1 processing

Micrographs were CTF-corrected using CTFFIND4 (*Rohou and Grigorieff, 2015*) in Relion (*Scheres, 2012*). A small subset of micrographs was used to generate initial 2D averages using the Relion Laplacian autopicker, which were then used as a template on the total dataset for template-based particle picking, resulting in 278,234 total particles. After a round of 2D classification, 155,574 particles were selected. The 3D classification scheme is detailed in *Figure 3—figure supplement 2*. Both CtfRefinement and Bayesian polishing were applied to the final 38,908 particle set, and then refined with D5 symmetry (a decamer of stacked pentamers) and sharpened using postprocessing (*Scheres, 2012*). The final structure was refined to an estimated 3.60 Å.

A homology model of BFLF1 was generated using the ORF68 atomic model in SWISS MODEL (*Waterhouse et al., 2018*). Subsequent refinement was done with iterative rounds of manual model building in *Coot* (*Emsley et al., 2010*) and automated refinement using torsion-angle NCS restraints in phenix.refine (*Adams et al., 2010*; *Afonine et al., 2012*; *Liebschner et al., 2019*). Data collection and refinement statistics are listed in *Supplementary file 1*.

## Crystallization and structure determination

Crystals of ORF68 were obtained by hanging drop vapor diffusion with 2 µL of concentrated protein (20 mg/mL) and 2 µL of crystallization solution (310 mM $CaCl_2$, 95 mM HEPES pH 7.5, 26.6% PEG-400, 5% glycerol) with equilibration against 1 mL of crystallization solution at 20°C. Crystals grew over the course of 1–3 days and were harvested and flash-cooled in liquid nitrogen without additional cryoprotection. X-ray diffraction data were collected at 100 K on a single crystal at beamline 8.3.1 at the Advanced Light Source (Berkeley, CA). Data were integrated with XDS (*Kabsch, 2010*), and the space group was determined using *POINTLESS* (*Evans, 2011*). Data were merged and

elliptically truncated to correct for anisotropy using the STARANISO server (*Tickle et al., 2018*). The diffraction limits along the a*, b*, and c* axes were 2.86, 2.60, and 2.21 Å, respectively. The corrected anisotropy amplitudes were used for molecular replacement in PHASER (*McCoy et al., 2007*) using the partially built cryo-EM model of ORF68. Subsequent refinement was done with iterative rounds of manual model building in *Coot* (*Emsley et al., 2010*) and automated refinement with TLS and torsion-angle NCS restraints in phenix.refine (*Adams et al., 2010*; *Afonine et al., 2012*; *Liebschner et al., 2019*). Data collection and refinement statistics are listed in *Supplementary file 2*.

All figures were generated with PyMol (http://www.pymol.org). The electrostatic surface was calculated using APBS (*Baker et al., 2001*) in PyMol. Surface conservation as depicted in *Figure 3—figure supplement 1* was generated using the ConSurf server (*Ashkenazy et al., 2016*).

## Sequence alignment

The sequence of ORF68 was used for a BLAST (NCBI) search to find homologs in the *Herpesviridae*. No clear homologs were readily identified in the *Herpesvirales* families *Alloherpesviridae* and *Malacoherpesviridae*. A multiple sequence alignment (*Figure 1—figure supplement 3*) was generated using Clustal Omega (*Madeira et al., 2019*) and manually edited to condense long insertions relative to ORF68.

## Cell lines

HEK293T cells were maintained in Dulbecco's Modified Eagle Medium (DMEM) supplemented with 10% fetal bovine serum (FBS) (Seradigm). HEK293T cells constitutively expressing ORF68 (HEK293T-ORF68) were previously described (*Gardner and Glaunsinger, 2018*) and were maintained in DMEM supplemented with 10% FBS and 500 µg/mL zeocin. iSLK-puro cells were maintained in DMEM supplemented with 10% FBS and 1 µg/mL puromycin. The iSLK-BAC16 system consists of the KSHV genome on a bacterial artificial chromosome (BAC16) and a doxycycline-inducible copy of the KSHV lytic transactivator RTA (*Brulois et al., 2012*; *Myoung and Ganem, 2011*; *Stürzl et al., 2013*). All iSLK-BAC16 cell lines were maintained in DMEM supplemented with 10% FBS, 1 mg/mL hygromycin, and 1 µg/mL puromycin. iSLK-BAC16-ORF68.stop cells were complemented with pLJM1-2xStrep- or untagged ORF68 wild-type, mutants, or homologs by lentiviral transduction as described below and were maintained in DMEM supplemented with 10% FBS, 1 mg/mL hygromycin, 1 µg/mL puromycin, and 500 µg/mL zeocin. HEK293Ts were authenticated using short tandem repeat analysis. iSLK cells were derived from SLK cells (*Stürzl et al., 2013*), which are a renal cell carcinoma line listed on the International Cell Line Authentication Committee register of commonly misidentified cell lines, but which are one of the few cell lines that can be latently infected with a recombinant KSHV BAC and efficiently reactivated. Cell lines were tested for mycoplasma contamination by PCR and immunofluorescence and found to be mycoplasma-negative. Details for cell lines used in this study can also be found in the Key Resources Table (Appendix).

## Cell line establishment and viral mutagenesis

Complemented iSLK-BAC16-ORF68.stop cells (*Gardner and Glaunsinger, 2018*) were generated by lentiviral transduction. Lentivirus was generated in HEK293T cells by co-transfection of pLJM1-ORF68 wild-type, mutant, or a homolog along with the packaging plasmids pMD2.G and psPAX2. After 48 hr, the supernatant was harvested and syringe-filtered through a 0.45 µm filter. The supernatant was diluted 1:2 with DMEM, and polybrene was added to a final concentration of 8 µg/mL. Freshly trypsinized iSLK-BAC16-ORF68.stop cells ($1 \times 10^6$) were spinfected in a six-well plate for 2 hr at $876 \times g$. After 24 hr the cells were expanded to a 10 cm tissue culture plate and selected for 2 weeks in DMEM supplemented with 10% FBS, 1 mg/mL hygromycin, 1 µg/mL puromycin, and 500 µg/mL zeocin.

All viral ORF68 mutants were generated using the scarless Red recombination system in BAC16 GS1783 *E. coli* as previously described (*Brulois et al., 2012*). Modified BACs were purified using a Nucleobond BAC 100 kit (Clontech). BAC quality was assessed by digestion with RsrII and SbfI (New England Biolabs). Latently infected iSLK cell lines with modified virus were generated by transfection of HEK293T-ORF68 cells with 5 µg BAC DNA using PolyJet reagent (SignaGen). The following day transfected HEK293T cells were trypsinized and mixed 1:1 with freshly trypsinized iSLK-puro cells

and treated with 30 nM 12-O-tetradecanoylphorbyl-13-acetate and 300 μM sodium butyrate for 4 days to induce lytic replication. iSLK cells were then selected in medium containing 300 μg/mL hygromycin B, 1 μg/mL puromycin, and 250 μg/mL G418. The hygromycin B concentration was increased to 500 μg/mL and 1 mg/mL until all HEK293T cells died.

## Virus characterization

For reactivation studies, $1 \times 10^6$ iSLK cells or iSLK.ORF68.stop cells complemented with wild-type or mutant ORF68 were plated in 10 cm dishes for 16 hr, then reactivated with 5 μg/mL doxycycline and 1 mM sodium butyrate for an additional 72 hr or left unreactivated. Infectious virion production was determined by supernatant transfer assay. Supernatant from reactivated iSLK cells was syringe-filtered through a 0.45 μm filter, then 2 mL of supernatant was spinoculated onto $1 \times 10^6$ freshly trypsinized HEK293T cells for 2 hr at $876 \times g$. After 24 hr, the media was aspirated, and the cells were washed with cold PBS and crosslinked in 4% paraformaldehyde (Electron Microscopy Services) diluted in PBS. Cells were pelleted at $1000 \times g$ for 5 min at 4°C, resuspended in PBS, and 50,000 cells/sample were analyzed on a BD Accuri C6 flow cytometer. The data were analyzed using FlowJo version 10 (BD Biosciences).

To determine fold DNA replication, reactivated and unreactivated iSLK cells were rinsed with PBS, scraped, pelleted at $1000 \times g$ for 5 min at 4°C, and stored at −80°C. Cells were resuspended in 600 μL of PBS, of which 200 μL was purified using a NucleoSpin Blood kit (Macherey Nagel) according to the manufacturer's instructions. The fold DNA replication was quantified by qPCR using iTaq Universal SYBR Green Supermix (Bio-Rad) on a QuantStudio3 Real-Time PCR machine. DNA levels were quantified with primers specific for the KSHV ORF57 promoter and normalized to human CTGF promoter and to unreactivated samples to determine fold replication (*Supplementary file 3*).

Total protein was isolated from reactivated iSLK cells at 72 hr. Samples were resuspended in lysis buffer, rotated for 30 min at 4°C, and clarified by centrifugation at $21,000 \times g$ for 10 min at 4°C. Lysate (25 μg) was used for SDS-PAGE and western blotting in Tris-buffered saline and 0.2% Tween 20 (TBST) using rabbit anti-K8.1 (1:10,000), rabbit anti-ORF68 (1:5000), rabbit anti-ORF6 (1:10,000), and mouse anti-GAPDH (1:5000; Abcam). Rabbit anti-K8.1, anti-ORF6, and anti-ORF68 antibodies were previously described (*Gardner and Glaunsinger, 2018*; *Didychuk et al., 2020*). Following incubation with primary antibodies, membranes were washed with TBST and incubated with goat anti-rabbit-HRP (1:5000; Southern Biotech) or goat anti-mouse-HRP (1:5000; Southern Biotech).

Southern blotting iSLK-BAC16 cells ($1 \times 10^6$) were harvested after 72 hr of reactivation. Cells were rinsed with PBS, scraped, pelleted at $1000 \times g$ for 5 min at 4°C, and stored at −80°C. Cells were resuspended in 700 μL of Hirt lysis buffer (10 mM Tris-HCl pH 7.4, 10 mM EDTA, 0.6% SDS) and incubated at room temperature for 5 min. NaCl was added to a final concentration of 0.85 M, and samples were rotated at 4°C overnight. The following day insoluble material was pelleted by centrifugation at $21,000 \times g$ for 30 min at 4°C. The supernatant was then treated with 100 μg/mL RNase A (ThermoFisher) for 1 hr at 55°C, then with 200 μg/mL proteinase K (Promega) for 1 hr at 55°C. DNA was isolated by phenol-chloroform extraction and ethanol precipitation. DNA (5 μg) was digested with PstI-HF (New England Biolabs) overnight, then separated by electrophoresis on a 0.7% agarose 1× TBE gel. The gel was denatured in 0.5 M NaOH, 1.5 M NaCl, neutralized in 1 M Tris pH 7.4, 1.5 M NaCl, then transferred to an Amersham Hybond-N+ membrane (GE Healthcare) by capillary action in 20× SSC (3 M NaCl, 0.3 M sodium citrate pH 7.0) overnight, and cross-linked to the membrane in a StrataLinker 2400 (Stratagene) using the AutoUV setting. The membrane was treated according to the DIG High Prime DNA Labeling and Detection Starter Kit II (Roche) according to the manufacturer's instructions using a DIG-labeled DNA probe corresponding to a single repeat of the KSHV terminal repeats as previously described (*Gardner and Glaunsinger, 2018*). After overnight hybridization, washing, blocking, and incubation with anti-DIG-AP antibody, the membrane was visualized on a ChemiDoc MP imaging system (Bio-Rad).

## Electrophoretic mobility shift assays

Fluorescein-labeled dsDNA probes (*Supplementary file 3*) (Integrated DNA Technologies) were prepared as 2× stocks (20 nM) in binding buffer (100 mM NaCl, 20 mM HEPES pH 7.5, 5% glycerol, 0.05% CHAPS). Wild-type or mutant ORF68 (retaining the 2xStrep tag and HRV 3C protease cleavage site) was diluted in binding buffer containing 0.2 mg/mL Bovine Serum Albumin (BSA). Final

binding reactions were prepared by mixing equal volumes of probe DNA and protein, and thus contained 10 nM DNA probe, 100 mM NaCl, 20 mM HEPES pH 7.5, 5% glycerol, 0.05% CHAPS, 0.1 mg/mL BSA, and variable concentrations of ORF68. Concentrations listed are for the monomer. Samples were incubated at room temperature for 20 min prior to electrophoresis on a 5% polyacrylamide (29:1 acrylamide:bis-acrylamide)/1× Tris borate gel at 2W at 4°C. Gels were imaged using a ChemiDoc MP imaging system (Bio-rad). Results were analyzed using Fiji (*Schindelin et al., 2012*). The percent bound probe was determined by dividing the intensity of the shifted band by the total intensity of the lane. Binding curves were generated using nonlinear regression in GraphPad Prism 8 to the following equation: % bound = $(B_{max} *[protein]^H)/(EC50^H + [protein]^H)$. We report the effective binding affinity (EC50) rather than binding affinity ($K_d$) because the concentration of fluorescent probe used in our assays and relatively high affinity of ORF68 for DNA preclude use of the standard Hill–Langmuir equation, which does not incorporate ligand depletion.

## Acknowledgements

We thank current and former members of the Glaunsinger and Martin labs, particularly Angelica Castañeda, Divya Nandakumar, Jessica Tucker, and Chloe McCollum, for their helpful suggestions. We are grateful to Eric Montemayor for advice. We thank beamline staff at ALS 8.2.2 and 8.3.1 for assistance with data collection, and UC Berkeley Cal-Cryo Facility members for assistance with cryo-EM data collection. AD is the Rhee Family Fellow of the Damon Runyon Cancer Research Foundation (DRG-2349–18). SG is a Howard Hughes Medical Institute Fellow of the Damon Runyon Cancer Research Foundation (DRG-2342-18). BG and AM are investigators of the Howard Hughes Medical Institute. This research was supported by NIH R01AI122528 to BG Beamlines 8.2.2 and 8.3.1 of the Advanced Light Source, a U.S. DOE Office of Science User Facility under Contract No. DE-AC02-05CH11231, are supported in part by the ALS-ENABLE program funded by the National Institutes of Health, National Institute of General Medical Sciences, grant P30 GM124169-01.

## Additional information

### Competing interests

Andreas Martin: Reviewing editor, *eLife*. The other authors declare that no competing interests exist.

### Funding

| Funder | Grant reference number | Author |
| --- | --- | --- |
| Damon Runyon Cancer Research Foundation | DRG-2349-18 | Allison L Didychuk |
| Damon Runyon Cancer Research Foundation | DRG-2342-18 | Stephanie N Gates |
| Howard Hughes Medical Institute | | Andreas Martin<br>Britt A Glaunsinger |
| National Institutes of Health | R01AI122528 | Britt A Glaunsinger |

The funders had no role in study design, data collection and interpretation, or the decision to submit the work for publication.

### Author contributions

Allison L Didychuk, Conceptualization, Formal analysis, Funding acquisition, Investigation, Methodology, Writing - original draft, Writing - review and editing; Stephanie N Gates, Formal analysis, Funding acquisition, Investigation, Methodology, Writing - review and editing; Matthew R Gardner, Lisa M Strong, Resources, Investigation; Andreas Martin, Britt A Glaunsinger, Supervision, Funding acquisition, Writing - review and editing

Author ORCIDs
Allison L Didychuk ⓘ https://orcid.org/0000-0001-7277-5233
Stephanie N Gates ⓘ https://orcid.org/0000-0002-4312-2900
Matthew R Gardner ⓘ https://orcid.org/0000-0003-1334-5879
Lisa M Strong ⓘ https://orcid.org/0000-0002-4293-8131
Andreas Martin ⓘ http://orcid.org/0000-0003-0923-3284
Britt A Glaunsinger ⓘ https://orcid.org/0000-0003-0479-9377

### Decision letter and Author response
Decision letter https://doi.org/10.7554/eLife.62261.sa1
Author response https://doi.org/10.7554/eLife.62261.sa2

## Additional files

### Supplementary files
• Supplementary file 1. Cryo-EM data collection statistics. The cryo-EM maps for ORF68 and BFLF1 and the coordinate set for BFLF1 are available in *Supplementary file 4*.

• Supplementary file 2. X-ray data collection and refinement statistics for ORF68. Statistics for the highest-resolution shell are shown in parentheses. The STARANISO server was used for ellipsoidal truncation (*Tickle et al., 2018*). The worst diffraction limit after cutoff was 2.99 Å. The ellipsoidally truncated data set was deposited in the Protein Data Bank and is available in *Supplementary file 4*. Merged diffraction data that has not been ellipsoidally truncated is also available in *Supplementary file 4*. The coordinate set deposited in the Protein Data Bank is available as *Supplementary file 4*.

• Supplementary file 3. Oligonucleotides used for cloning, qPCR, and EMSAs.

• Supplementary file 4. Diffraction data sets, cryo-EM maps, and coordinate sets for ORF68 and BFLF1.

• Transparent reporting form

### Data availability
Atomic coordinates and structure factors for ORF68 have been deposited in the Protein Data Bank with accession code 6XF9. Diffraction images have been deposited in the SBGrid Data Bank under ID 794 (https://doi:10.15785/SBGRID/794). Cryo-EM maps for ORF68 and BFLF1 have been deposited in the Electron Microscopy Data Bank with accession codes EMD-22167 and EMD-22168. The atomic model of BFLF1 was deposited in the Protein Data Bank with accession code 6XFA. Final coordinate sets, structure factors with calculated phases, and cryo-EM maps are provided as Supplementary File 4.

The following datasets were generated:

| Author(s) | Year | Dataset title | Dataset URL | Database and Identifier |
|---|---|---|---|---|
| Didychuk AL, Gates SN, Gardner MR, Strong LM, Martin A, Glaunsinger BA | 2020 | Crystal structure of KSHV ORF68 | https://www.rcsb.org/structure/6XF9 | RCSB Protein Data Bank, 6XF9 |
| Didychuk AL, Gates SN, Gardner MR, Strong LM, Martin A, Glaunsinger BA | 2020 | X-Ray Diffraction data from KSHV ORF68, source of 6XF9 structure | https://doi.org/10.15785/SBGRID/794 | SBGrid Data Bank, 10.15785/SBGRID/794 |
| Didychuk AL, Gates SN, Gardner MR, Strong LM, Martin A, Glaunsinger BA | 2020 | Cryo-EM map of EBV BFLF1 | http://www.ebi.ac.uk/pdbe/entry/emdb/EMD-22168 | Electron Microscopy Data Bank, EMD-22168 |
| Didychuk AL, Gates SN, Gardner MR, Strong LM, Martin | 2020 | Cryo-EM map of KSHV ORF68 | http://www.ebi.ac.uk/pdbe/entry/emdb/EMD-22167 | Electron Microscopy Data Bank, EMD-22167 |

A, Glaunsinger BA

| Didychuk AL, Gates SN, Gardner MR, Strong LM, Martin A, Glaunsinger BA | 2020 | Cryo-EM structure of EBV BFLF1 | https://www.rcsb.org/structure/6XFA | RCSB Protein Data Bank, 6XFA |

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

# Appendix 1

**Appendix 1—key resources table**

| Reagent type (species) or resource | Designation | Source or reference | Identifiers | Additional information |
|---|---|---|---|---|
| Gene (*Kaposi's sarcoma-associated herpesvirus*) | ORF68 | Human gammaherpesvirus 8 clone BAC16, complete genome | GenBank: MK733609.1; Uniprot: D2XQF2 | |
| Gene (*Epstein–Barr virus*) | BFLF1 | Human gammaherpesvirus 4 isolate NPCT090, complete genome | GenBank: MK540447.1 | |
| Gene (*MHV68*) | mu68 | Murid herpesvirus 4 strain g2.4, complete genome | GenBank: AF105037.1 Uniprot: O41969 | |
| Gene (*human cytomegalovirus*) | UL52 | Human herpesvirus 5 strain Towne, complete genome | GenBank: FJ616285.1; Uniprot: O56765 | |
| Gene (*Herpes simplex virus 1*) | UL32 | Human herpesvirus 1 isolate KOS, complete genome | GenBank: KT899744.1; Uniprot: H9E939 | |
| Strain, strain background (*Escherichia coli*) | GS1783 | ***Brulois et al., 2012*** | PMID: 22740391 | WT BAC16-containing *E. coli* used for construction of BAC16 mutants |
| Genetic reagent (*Kaposi's sarcoma-associated herpesvirus*) | KSHV BAC16 WT | ***Brulois et al., 2012*** | PMID: 22740391 | WT BAC16 containing KSHV genome |
| Genetic reagent (*Kaposi's sarcoma-associated herpesvirus*) | KSHV BAC16 ORF68.stop (68S) | ***Gardner and Glaunsinger, 2018*** | PMID: 29875246 | BAC16 mutant containing a premature stop codon in ORF68 |
| Genetic reagent (*Kaposi's sarcoma-associated herpesvirus*) | KSHV BAC16 ORF68.stop-MR (68S-MR) | ***Gardner and Glaunsinger, 2018*** | PMID: 29875246 | Mutant rescue of BAC16 ORF68.stop |
| Genetic reagent (*Kaposi's sarcoma-associated herpesvirus*) | KSHV BAC16 ORF68-K435A | This paper; 'Materials and methods' (Cell line establishment and viral mutagenesis) | | BAC16 mutant containing ORF68-K435A |
| Genetic reagent (*Kaposi's sarcoma-associated herpesvirus*) | KSHV BAC16 ORF68-K435A-MR | This paper; 'Materials and methods' (Cell line establishment and viral mutagenesis) | | Mutant rescue of BAC16 ORF68-K435A |
| Genetic reagent (*Kaposi's sarcoma-associated herpesvirus*) | KSHV BAC16 ORF68-K174A/R179A/K182A (3+) | This paper; 'Materials and methods' (Cell line establishment and viral mutagenesis) | | BAC16 mutant containing ORF68-K174A/R179A/K182A |
| Genetic reagent (*Kaposi's sarcoma-associated herpesvirus*) | KSHV BAC16 K174A/R179A/K182A-MR (3+-MR) | This paper; 'Materials and methods' (Cell line establishment and viral mutagenesis) | | Mutant rescue of BAC16 ORF68-K174A/R179A/K182A |
| Genetic reagent (*Kaposi's sarcoma-associated herpesvirus*) | KSHV BAC16 ORF68-TS | This paper; 'Materials and methods' (Cell line establishment and viral mutagenesis) | | BAC16 mutant containing ORF68-TS |

*Continued on next page*

*Appendix 1—key resources table continued*

| Reagent type (species) or resource | Designation | Source or reference | Identifiers | Additional information |
|---|---|---|---|---|
| Genetic reagent (*Kaposi's sarcoma-associated herpesvirus*) | KSHV BAC16 ORF68-TS-MR | This paper; 'Materials and methods' (Cell line establishment and viral mutagenesis) | | BAC16 mutant containing ORF68-TS-MR |
| Cell line (*Homo sapiens*) | HEK-293T | ATCC | CRL-3216 | This cell line is commercially available from ATCC |
| Cell line (*Homo sapiens*) | iSLK-puro | *Myoung and Ganem, 2011*; *Stürzl et al., 2013* | PMID: 21419799; PMID: 22987579 | Renal-cell carcinoma cell line containing doxycycline-inducible KSHV RTA |
| Cell line (*Homo sapiens, Kaposi's sarcoma-associated herpesvirus*) | iSLK-BAC16 WT | This paper; 'Materials and methods' (Cell line establishment and viral mutagenesis) | | iSLK-puro latently infected with KSHV derived from BAC16 WT |
| Cell line (*Homo sapiens, Kaposi's sarcoma-associated herpesvirus*) | iSLK-BAC16 ORF68.stop | This paper; 'Materials and methods' (Cell line establishment and viral mutagenesis) | | iSLK-puro latently infected with KSHV derived from BAC16 ORF68.stop |
| Cell line (*Homo sapiens, Kaposi's sarcoma-associated herpesvirus*) | iSLK-BAC16 ORF68.stop-MR | This paper; 'Materials and methods' (Cell line establishment and viral mutagenesis) | | iSLK-puro latently infected with KSHV derived from BAC16 ORF68.stop-MR |
| Cell line (*Homo sapiens, Kaposi's sarcoma-associated herpesvirus*) | iSLK-BAC16 ORF68-K435A | This paper; 'Materials and methods' (Cell line establishment and viral mutagenesis) | | iSLK-puro latently infected with KSHV derived from BAC16 ORF68-K435A |
| Cell line (*Homo sapiens, Kaposi's sarcoma-associated herpesvirus*) | iSLK-BAC16 ORF68-K435A-MR | This paper; 'Materials and methods' (Cell line establishment and viral mutagenesis) | | iSLK-puro latently infected with KSHV derived from BAC16 ORF68-K435A-MR |
| Cell line (*Homo sapiens, Kaposi's sarcoma-associated herpesvirus*) | iSLK-BAC16 ORF68-K174A/R179A/K182A | This paper; 'Materials and methods' (Cell line establishment and viral mutagenesis) | | iSLK-puro latently infected with KSHV derived from BAC16 ORF68- K174A/R179A/K182A |
| Cell line (*Homo sapiens, Kaposi's sarcoma-associated herpesvirus*) | iSLK-BAC16 ORF68- K174A/R179A/K182A -MR | This paper; 'Materials and methods' (Cell line establishment and viral mutagenesis) | | iSLK-puro latently infected with KSHV derived from BAC16 ORF68- K174A/R179A/K182A -MR |
| Cell line (*Homo sapiens, Kaposi's sarcoma-associated herpesvirus*) | iSLK-BAC16 ORF68-TS | This paper; 'Materials and methods' (Cell line establishment and viral mutagenesis) | | iSLK-puro latently infected with KSHV derived from BAC16 ORF68-TS |
| Cell line (*Homo sapiens, Kaposi's sarcoma-associated herpesvirus*) | iSLK-BAC16 ORF68-TS-MR | This paper; 'Materials and methods' (Cell line establishment and viral mutagenesis) | | iSLK-puro latently infected with KSHV derived from BAC16 ORF68-TS-MR |

*Continued on next page*

*Appendix 1—key resources table continued*

| Reagent type (species) or resource | Designation | Source or reference | Identifiers | Additional information |
|---|---|---|---|---|
| Cell line (*Homo sapiens*, *Kaposi's sarcoma-associated herpesvirus*) | iSLK-BAC16 ORF68.stop + pLJM1-zeo-empty | This paper; 'Materials and methods' (Cell line establishment and viral mutagenesis) | | Lentivirally transduced iSLK-BAC16 ORF68.stop |
| Cell line (*Homo sapiens*, *Kaposi's sarcoma-associated herpesvirus*) | iSLK-BAC16 ORF68.stop + pLJM1-zeo-2xStrep-ORF68 | This paper; 'Materials and methods' (Cell line establishment and viral mutagenesis) | | Lentivirally transduced iSLK-BAC16 ORF68.stop |
| Cell line (*Homo sapiens*, *Kaposi's sarcoma-associated herpesvirus*) | iSLK-BAC16 ORF68.stop + pLJM1-zeo-2xStrep-ORF68-C52A | This paper; 'Materials and methods' (Cell line establishment and viral mutagenesis) | | Lentivirally transduced iSLK-BAC16 ORF68.stop |
| Cell line (*Homo sapiens*, *Kaposi's sarcoma-associated herpesvirus*) | iSLK-BAC16 ORF68.stop + pLJM1-zeo-2xStrep-mu68 | This paper; 'Materials and methods' (Cell line establishment and viral mutagenesis) | | Lentivirally transduced iSLK-BAC16 ORF68.stop |
| Cell line (*Homo sapiens*, *Kaposi's sarcoma-associated herpesvirus*) | iSLK-BAC16 ORF68.stop + pLJM1-zeo-2xStrep-BFLF1 | This paper; 'Materials and methods' (Cell line establishment and viral mutagenesis) | | Lentivirally transduced iSLK-BAC16 ORF68.stop |
| Cell line (*Homo sapiens*, *Kaposi's sarcoma-associated herpesvirus*) | iSLK-BAC16 ORF68.stop + pLJM1-zeo-2xStrep-UL52 | This paper; 'Materials and methods' (Cell line establishment and viral mutagenesis) | | Lentivirally transduced iSLK-BAC16 ORF68.stop |
| Cell line (*Homo sapiens*, *Kaposi's sarcoma-associated herpesvirus*) | iSLK-BAC16 ORF68.stop + pLJM1-zeo-UL32 | This paper; 'Materials and methods' (Cell line establishment and viral mutagenesis) | | Lentivirally transduced iSLK-BAC16 ORF68.stop |
| Antibody | Anti-vinculin (rabbit polyclonal) | Abcam | Cat. no. ab91459 | WB primary (1:1000) |
| Antibody | Strep-Tag II HRP conjugate | Novagen | Cat. no. 71591-3 | WB primary (1:2500) |
| Antibody | Anti-K8.1 (rabbit polyclonal) | *Gardner and Glaunsinger, 2018* | PMID: 29875246 | WB primary (1:10,000) |
| Antibody | Anti-ORF68 (rabbit polyclonal) | *Gardner and Glaunsinger, 2018* | PMID: 29875246 | WB primary (1:5000) |
| Antibody | Anti-ORF6 (Rabbit polyclonal) | *Gardner and Glaunsinger, 2018* | PMID: 29875246 | WB primary (1:10,000) |
| Antibody | Anti-GAPDH (mouse monoclonal) | Abcam | Cat. no. ab8245 | WB primary (1:5000) |
| Antibody | Anti-rabbit IgG-HRP (goat polyclonal) | Southern Biotech | Cat. no. 4030-05 | WB secondary (1:5000) |
| Antibody | Anti-mouse IgG-HRP (goat polyclonal) | Southern Biotech | Cat. no. 1031-05 | WB secondary (1:5000) |

*Continued on next page*

*Appendix 1—key resources table continued*

| Reagent type (species) or resource | Designation | Source or reference | Identifiers | Additional information |
|---|---|---|---|---|
| Recombinant DNA reagent | pcDNA4/TO-2xStrep-ORF68 | This paper; 'Materials and methods' (Plasmids) | Addgene: 162625 | For transient expression of ORF68 |
| Recombinant DNA reagent | pcDNA4/TO-ORF68-2xStrep | *Davis et al., 2015* | Addgene: 136229 | For transient expression of ORF68 |
| Recombinant DNA reagent | pcDNA4/TO-ORF68 (tagless) | This paper; 'Materials and methods' (Plasmids) | Addgene: 166025 | For transient expression of ORF68 |
| Recombinant DNA reagent | pcDNA4/TO-2xStrep-BFLF1 | This paper; 'Materials and methods' (Plasmids) | Addgene: 162626 | For transient expression of BFLF1 |
| Recombinant DNA reagent | pcDNA4/TO-2xStrep-mu68 | This paper; 'Materials and methods' (Plasmids) | Addgene: 162627 | For transient expression of mu68 |
| Recombinant DNA reagent | pcDNA4/TO-2xStrep-UL52 | This paper; 'Materials and methods' (Plasmids) | Addgene: 162628 | For transient expression of UL52 |
| Recombinant DNA reagent | pcDNA4/TO-2xStrep-UL32 | This paper; 'Materials and methods' (Plasmids) | Addgene: 162629 | For transient expression of UL32 |
| Recombinant DNA reagent | pcDNA4/TO-2xStrep-ORF68 C52A | This paper; 'Materials and methods' (Plasmids) | Addgene: 162630 | For transient expression of ORF68 C52A |
| Recombinant DNA reagent | pcDNA4/TO-2xStrep-ORF68 C373A | This paper; 'Materials and methods' (Plasmids) | Addgene: 162631 | For transient expression of ORF68 C373A |
| Recombinant DNA reagent | pcDNA4/TO-2xStrep-ORF68 C191A/C192A | This paper; 'Materials and methods' (Plasmids) | Addgene:162632 | For transient expression of ORF68 C191A/C192A |
| Recombinant DNA reagent | pcDNA4/TO-2xStrep-ORF68 C415A | This paper; 'Materials and methods' (Plasmids) | Addgene: 162633 | For transient expression of ORF68 C415A |
| Recombinant DNA reagent | pcDNA4/TO-2xStrep-ORF68 H452A | This paper; 'Materials and methods' (Plasmids) | Addgene: 162634 | For transient expression of ORF68 H452A |
| Recombinant DNA reagent | pcDNA4/TO-2xStrep-ORF68 C79A | This paper; 'Materials and methods' (Plasmids) | Addgene: 162635 | For transient expression of ORF68 C79A |
| Recombinant DNA reagent | pcDNA4/TO-2xStrep-ORF68 K435A | This paper; 'Materials and methods' (Plasmids) | Addgene: 162636 | For transient expression of ORF68 K435A |
| Recombinant DNA reagent | pcDNA4/TO-2xStrep-ORF68 R443A | This paper; 'Materials and methods' (Plasmids) | Addgene: 162637 | For transient expression of ORF68 R443A |
| Recombinant DNA reagent | pcDNA4/TO-2xStrep-ORF68 K450A/R451A | This paper; 'Materials and methods' (Plasmids) | Addgene: 162638 | For transient expression of ORF68 K450A/R451A |
| Recombinant DNA reagent | pcDNA4/TO-2xStrep-ORF68 K174A/R179A/K182A | This paper; 'Materials and methods' (Plasmids) | Addgene: 162639 | For transient expression of ORF68 K174A/R179A/K182A |
| Recombinant DNA reagent | pcDNA4/TO-2xStrep-ORF68 R14A | This paper; 'Materials and methods' (Plasmids) | Addgene: 162640 | For transient expression of ORF68 R14A |
| Recombinant DNA reagent | pcDNA4/TO-2xStrep-ORF68 K310A | This paper; 'Materials and methods' (Plasmids) | Addgene: 162641 | For transient expression of ORF68 K310A |
| Recombinant DNA reagent | pcDNA4/TO-2xStrep-ORF68 R14A/K310A | This paper; 'Materials and methods' (Plasmids) | Addgene: 162642 | For transient expression of ORF68 R14A/K310A |

*Continued on next page*

*Appendix 1—key resources table continued*

| Reagent type (species) or resource | Designation | Source or reference | Identifiers | Additional information |
|---|---|---|---|---|
| Recombinant DNA reagent | pcDNA4/TO-2xStrep-ORF68 K395A/K396A | This paper; 'Materials and methods' (Plasmids) | Addgene: 162643 | For transient expression of ORF68 K395A/K396A |
| Recombinant DNA reagent | pcDNA4/TO-2xStrep-UL32 C128A | This paper; 'Materials and methods' (Plasmids) | Addgene: 162644 | For transient expression of UL32 C128A |
| Recombinant DNA reagent | pcDNA4/TO-2xStrep-UL32 C502A | This paper; 'Materials and methods' (Plasmids) | Addgene: 162645 | For transient expression of UL32 C502A |
| Recombinant DNA reagent | pcDNA4/TO-2xStrep-UL32 C308A/C309A | This paper; 'Materials and methods' (Plasmids) | Addgene: 162646 | For transient expression of UL32 C308A/C309A |
| Recombinant DNA reagent | pcDNA4/TO-2xStrep-UL32 C544A | This paper; 'Materials and methods' (Plasmids) | Addgene: 162647 | For transient expression of UL32 C544A |
| Recombinant DNA reagent | pcDNA4/TO-2xStrep-UL32 H581A | This paper; 'Materials and methods' (Plasmids) | Addgene: 162648 | For transient expression of UL32 H581A |
| Recombinant DNA reagent | pcDNA4/TO-2xStrep-UL32 C155A | This paper; 'Materials and methods' (Plasmids) | Addgene: 162649 | For transient expression of UL32 C155A |
| Recombinant DNA reagent | pUE1-TSP-ORF68 | This paper; 'Materials and methods' (Plasmids) | Addgene: 162650 | For transient overexpression of ORF68 |
| Recombinant DNA reagent | pUE1-TSP-ORF68 K435A | This paper; 'Materials and methods' (Plasmids) | Addgene: 162651 | For transient overexpression of ORF68 K435A |
| Recombinant DNA reagent | pUE1-TSP-ORF68 R443A | This paper; 'Materials and methods' (Plasmids) | Addgene: 162652 | For transient overexpression of ORF68 R443A |
| Recombinant DNA reagent | pUE1-TSP-ORF68 K450A/R451A | This paper; 'Materials and methods' (Plasmids) | Addgene: 162653 | For transient overexpression of ORF68 K450A/R451A |
| Recombinant DNA reagent | pUE1-TSP-ORF68 K174A/R179A/K182A | This paper; 'Materials and methods' (Plasmids) | Addgene: 162654 | For transient overexpression of ORF68 K174A/R179A/K182A |
| Recombinant DNA reagent | pUE1-TSP-ORF68 R14A/K310A | This paper; 'Materials and methods' (Plasmids) | Addgene: 162655 | For transient overexpression of ORF68 R14A/K310A |
| Recombinant DNA reagent | pUE1-TSP-ORF68 K395A/K396A | This paper; 'Materials and methods' (Plasmids) | Addgene: 162656 | For transient overexpression of ORF68 K395A/K396A |
| Recombinant DNA reagent | pUE1-TSP-BFLF1 | This paper; 'Materials and methods' (Plasmids) | Addgene: 162657 | For transient overexpression of BFLF1 |
| Recombinant DNA reagent | pUE1-TSP-UL32 | This paper; 'Materials and methods' (Plasmids) | Addgene: 162658 | For transient overexpression of UL32 |
| Recombinant DNA reagent | pLJM1-zeo-empty | *Gardner and Glaunsinger, 2018* | PMID: 29875246 | Empty lentiviral vector |
| Recombinant DNA reagent | pLJM1-zeo-2xStrep-ORF68 | This paper; 'Materials and methods' (Plasmids) | Addgene: 162659 | Lentiviral vector for stable expression of ORF68 |
| Recombinant DNA reagent | pLJM1-zeo-2xStrep-ORF68 C52A | This paper; 'Materials and methods' (Plasmids) | Addgene: 162660 | Lentiviral vector for stable expression of ORF68 C52A |
| Recombinant DNA reagent | pLJM1-zeo-2xStrep-BFLF1 | This paper; 'Materials and methods' (Plasmids) | Addgene: 162661 | Lentiviral vector for stable expression of BFLF1 |

*Continued on next page*

*Appendix 1—key resources table continued*

| Reagent type (species) or resource | Designation | Source or reference | Identifiers | Additional information |
|---|---|---|---|---|
| Recombinant DNA reagent | pLJM1-zeo-2xStrep-mu68 | This paper; 'Materials and methods' (Plasmids) | Addgene: 162662 | Lentiviral vector for stable expression of mu68 |
| Recombinant DNA reagent | pLJM1-zeo-2xStrep-UL52 | This paper; 'Materials and methods' (Plasmids) | Addgene: 162663 | Lentiviral vector for stable expression of UL52 |
| Recombinant DNA reagent | pLJM1-zeo-UL32 | This paper; 'Materials and methods' (Plasmids) | Addgene: 162664 | Lentiviral vector for stable expression of UL32 |
| Recombinant DNA reagent | psPAX2 | A gift from Didier Trono | Addgene: 12260 | Lentiviral packaging plasmid |
| Recombinant DNA reagent | pMD2.G | A gift from Didier Trono | Addgene: 12259 | Lentiviral packaging plasmid |
| Peptide, recombinant protein | HRV 3C protease | Millipore Sigma | Cat. no: 71493 | For removal of Twin-Strep tag during protein purification |
| Commercial assay or kit | In-Fusion HD Cloning kit | Clontech | Cat. no. 639650 | For cloning |
| Commercial assay or kit | NucleoBond BAC 100 kit | Macherey Nagel | Cat. no. 740579 | For preparation of BAC DNA |
| Commercial assay or kit | NucleoSpin Blood kit | Macherey Nagel | Cat. no. 740951.50 | For isolation of DNA from iSLK BAC16 cell lines |
| Commercial assay or kit | iTaq Universal SYBR Green Supermix | Bio-Rad | Cat. no. 1725122 | For qPCR assays |
| Commercial assay or kit | DIG High Prime DNA Labeling and Detection Starter Kit II | Roche | Cat. no. 11585614910 | For probe labeling and detection of Southern blots |
| Software, algorithm | CTTFFIND4 | PMID: 26278980 | RRID: SCR_016732 | |
| Software, algorithm | Gautomatch | K. Zhang (MRC-LMB (https://www2.mrc-lmb.cam.ac.uk/research/locally-developed-software/zhang-software/)) | | |
| Software, algorithm | Relion | PMID: 23000701 | RRID: SCR_016274 | |
| Software, algorithm | SerialEM | PMID: 16182563 | RRID: SCR_017293 | |
| Software, algorithm | Motioncor2 | PMID: 28250466 | RRID: SCR_016499 | |
| Software, algorithm | *Coot* | PMID: 20383002 | RRID: SCR_014222 | |
| Software, algorithm | phenix.refine | PMID: 20124702, 22505256, 31588918 | RRID: SCR_016736 | |
| Software, algorithm | XDS | PMID: 20124692 | RRID: SCR_015652 | |
| Software, algorithm | *POINTLESS* | PMID: 21460446 | RRID: SCR_014218 | |
| Software, algorithm | STARANISO server | STARANISO (staraniso.globalphasing.org) | RRID: SCR_018362 | |
| Software, algorithm | PHASER | PMID: 19461840 | RRID: SCR_014219 | |

*Continued on next page*

*Appendix 1—key resources table continued*

| Reagent type (species) or resource | Designation | Source or reference | Identifiers | Additional information |
|---|---|---|---|---|
| Software, algorithm | PyMol | PyMol (pymol.org) | RRID: SCR_000305 | |
| Software, algorithm | APBS | PMID: 11517324 | RRID: SCR_008387 | |
| Software, algorithm | ConSurf Server | PMID: 27166375 | RRID: SCR_002320 | |
| Software, algorithm | SWISS-MODEL | PMID: 29788355 | RRID: SCR_018123 | |
| Software, algorithm | Clustal Omega | PMID: 30976793 | RRID: SCR_001591 | |
| Software, algorithm | GraphPad Prism | GraphPad | RRID: SCR_002798 | Version 8 |
| Other | Strep-Tactin Superflow high-capacity 50% suspension | IBA Lifesciences | Cat. no. 2-4030-025 | |

