## [Decision Letter]

**Acceptance summary:**

The genome packaging machinery of herpesviruses is composed of 6 proteins. The functions of 5 of these have been relatively well characterized, but little is known about the 6th component, the conserved protein termed ORF68 in KSHV. Here, by obtaining a high-resolution structure of ORF68 (and its homolog from a closely related EBV), Didychuk et al. show that it forms a pentameric ring with a positively charged pore that could accommodate dsDNA. Authors further show that the basic residues lining the pore are essential for DNA binding, genome packaging, and viral replication. These data for the first time suggest that ORF68 binds the dsDNA genome and may, in some manner, act as an adaptor bringing the genome and the genome-packaging terminase motor to the capsid portal.

**Decision letter after peer review:**

Thank you for submitting your article "A pentameric protein ring with novel architecture is required for herpesviral packaging" for consideration by *eLife*. Your article has been reviewed by three peer reviewers, and the evaluation has been overseen by a Reviewing Editor and Cynthia Wolberger as the Senior Editor. The following individual involved in review of your submission has agreed to reveal their identity: Stephen C Graham (Reviewer #2).

The reviewers have discussed the reviews with one another and the Reviewing Editor has drafted this decision to help you prepare a revised submission.

Summary:

Didychuk et al. report crystal and cryo-EM structures of the ORF68 protein from KSHV/HHV-8, plus the cryo-EM structure of its homologue BFLF1 from EBV/HHV-4. These structures, along with biochemical data presented in this paper and the group's previous work, demonstrate convincingly that ORF68 is a DNA-binding protein involved in genome packaging. Importantly, the authors show that the conserved cysteine residues in ORF68 mediate zinc ligation, suggesting that they play a structural role rather than a role in intracellular disulfide bond regulation (as had been hypothesised for the HSV-1/HHV-1 homologue pUL32). The work is methodologically sound and provides a structural framework for probing the function of ORF68 and homologues in virus assembly.

Essential revisions:

1) The authors assert that ORF68, BFLF1 and UL32 all form pentamers, and that this is the active form of these proteins. While this is supported by the EM analysis of ORF68 and UL32, the assertion that BFLF1 is also most likely active as a pentamer (subsection “Homologs of ORF68 form similar structures”) is not supported by data. Ideally the authors would use analytical ultracentrifugation or MALS to define the oligomeric state of the particles in solution, but analytical size exclusion chromatography would be sufficient to confirm that ORF68, BFLF1 and UL32 all form similarly sized particles in solution.

2) The structural work presented in this manuscript show compellingly that ORF68 and BFLF1 share the same fold, and sequence conservation suggests that this fold will be conserved across alpha- and beta-herpesvirus homologues, UL32 and UL52 (respectively). However, building a homology model of UL32 and UL52 using ORF68 as a template structure does not provide additional support to this hypothesis – by definition a homology model will always look similar to its template structure. Figure 3C, D and discussion of the homology models should be removed in favour of a discussion of sequence conservation (Figure 1—figure supplement 3).

3) The authors use EMSAs to probe the affinity ORF68 for “cognate” (GC-rich) or scrambled DNA. While the similar binding affinity can be easily seen, the estimated dissociation constant (Kd) is likely significantly wrong because the Langmuir-Hill equation used by the authors does not take into account ligand depletion and the assumption that the [ORF68]total equals [ORF68]free is not valid when using nM concentrations of both fluorescent DNA probe and ORF68. The authors should either quote the effective binding affinity in their assay (EC50) or fit their data to a model that takes into account ligand depletion.

---

## [Author Response]

Essential revisions:1) The authors assert that ORF68, BFLF1 and UL32 all form pentamers, and that this is the active form of these proteins. While this is supported by the EM analysis of ORF68 and UL32, the assertion that BFLF1 is also most likely active as a pentamer (subsection “Homologs of ORF68 form similar structures”) is not supported by data. Ideally the authors would use analytical ultracentrifugation or MALS to define the oligomeric state of the particles in solution, but analytical size exclusion chromatography would be sufficient to confirm that ORF68, BFLF1 and UL32 all form similarly sized particles in solution.

We have revisited our size exclusion chromatography data and found that BFLF1 does indeed appear to be a decamer (or at least larger than a pentamer) in solution. We have included these data in Figure 3—figure supplement 2. As a result, we have changed our language when referring to the oligomerization state of ORF68 and its homologs. While they are all *capable* of forming pentamers, the relevant in vivo oligomerization state remains to be determined. We have added a section in the Discussion to address this.

We have also included size exclusion chromatography to show that a representative ORF68 variant with a mutation in the central channel (ORF68 K435A) maintains the pentameric architecture (Figure 4—figure supplement 2).

2) The structural work presented in this manuscript show compellingly that ORF68 and BFLF1 share the same fold, and sequence conservation suggests that this fold will be conserved across alpha- and beta-herpesvirus homologues, UL32 and UL52 (respectively). However, building a homology model of UL32 and UL52 using ORF68 as a template structure does not provide additional support to this hypothesis – by definition a homology model will always look similar to its template structure. Figure 3C, D and discussion of the homology models should be removed in favour of a discussion of sequence conservation (Figure 1—figure supplement 3).

We have removed Figure 3C and D and expanded the discussion focusing on the sequence conservation across the herpesviruses.

3) The authors use EMSAs to probe the affinity ORF68 for “cognate” (GC-rich) or scrambled DNA. While the similar binding affinity can be easily seen, the estimated dissociation constant (Kd) is likely significantly wrong because the Langmuir-Hill equation used by the authors does not take into account ligand depletion and the assumption that the [ORF68]total equals [ORF68]free is not valid when using nM concentrations of both fluorescent DNA probe and ORF68. The authors should either quote the effective binding affinity in their assay (EC50) or fit their data to a model that takes into account ligand depletion.

Thank you for pointing this out- we now refer to effective binding affinity instead of *K*_d_ and have added a note in the Materials and methods. Alas, this is the drawback of using fluorescent probes for EMSAs.